# Research on Soil Moisture Inversion Method for Canal Slope of the Middle Route Project of the South to North Water Transfer Based on GNSS-R and Deep Learning

Qingfeng Hu [1], Yifan Li [1], Wenkai Liu [1], Weiqiang Lu [1], Hongxin Hai [1], Peipei He [1], Xianlin Liu [1,2,*], Kaifeng Ma [1], Dantong Zhu [1], Peng Wang [1] and Yingchao Kou [1]

[1] College of Surveying and Geo-Informatics, North China University of Water Resources and Electric Power, Zhengzhou 450046, China; huqingfeng@ncwu.edu.cn (Q.H.); z20211151026@stu.ncwu.edu.cn (Y.L.); liuwenkai@ncwu.edu.cn (W.L.); z202210151111@stu.ncwu.edu.cn (W.L.); z202210151100@stu.ncwu.edu.cn (H.H.); hepei@ncwu.edu.cn (P.H.); makaifeng@ncwu.edu.cn (K.M.); zhudantong@ncwu.edu.cn (D.Z.); z20211151045@stu.ncwu.edu.cn (P.W.); z20201150995@stu.ncwu.edu.cn (Y.K.)

[2] Chinese Academy of Engineering, Beijing 100088, China

[*] Correspondence: liuxl@cae.cn

**Abstract:** The soil moisture from the South-to-North Water Diversion Middle Route Project is assessed in this study. Complex and variable geological conditions complicate the prediction of soil moisture in the study area. To achieve this aim, we carried out research on soil moisture inversion methods for channel slopes in the study area using massive monitoring data from multiple GNSS observatories on channel slopes, incorporating GNSS-R techniques and deep learning algorithms. To address the issue of low accuracy in linear inversion when using a single satellite, this study proposes a multi-satellite and multi-frequency data fusion technique. Furthermore, three soil moisture inversion models, namely, the linear model, BP neural network model, and GA-BP neural network model, are established by incorporating deep learning techniques. In comparison with single-satellite data inversion, with the data fusion technique proposed in this study, the correlation is improved by 12.7%, the root mean square error is reduced by 0.217, the mean square error is decreased by 0.884, and the mean absolute error is decreased by 0.243 with the linear model. With the BP neural network model, the correlation is increased by 15.4%, the root mean square error is decreased by 0.395, the mean square error is decreased by 0.465, and the mean absolute error is reduced by 0.353. Moreover, with the GA-BP neural network model, the correlation is improved by 6.3%, the root mean square error is decreased by 1.207, the mean square error is decreased by 0.196, and the mean absolute error is reduced by 0.155. The results indicate that performing data fusion by using multiple satellites and multi-frequency bands is a feasible approach for improving the accuracy of soil moisture inversion. These research findings provide new technical means for the risk analysis of deformation disasters in the expansive soil channel slopes of the South-to-North Water Diversion Middle Route Project.

**Keywords:** data fusion; deep learning; drought detection; GNSS-R; soil moisture; South to North Water Transfer



## 1. Introduction

To address the issue of the uneven spatial distribution of water resources, as there are more water resources in the southern regions of China and less in the northern regions, the Chinese government has planned and implemented the South-to-North Water Diversion Middle Route Project. The South-to-North Water Diversion Middle Route Project is one of the world's major water conservancy projects. Its objective is to address the water scarcity issue in Northern China by diverting the abundant water resources from the southern regions of the country to the water-deficient areas in the north. The South-to-North Water Diversion Middle Route Project in China has a total length of 1432 km. The

project encounters complex and variable geological conditions, especially in the multiple sections along the channel that have expansive soil foundations [1]. Due to its characteristics of desiccation-induced consolidation and water-induced expansion, expansive soil is prone to the induction of deformation disasters in channel slopes during prolonged cycles of consolidation and expansion [2]. Therefore, the effective detection of soil moisture is crucial for ensuring the safe operation of this project. Soil moisture is one of the key factors influencing the stability of expansive soil channel slopes. By monitoring soil moisture, real-time information about soil water content can be obtained. This provides the support of scientific data for a comprehensive assessment of the risk of deformation disaster in the expansive soil channel slopes. The early detection of potential slope instability issues and the implementation of appropriate maintenance and repair measures can be facilitated to ensure the safety of the project [3]. Therefore, conducting research on soil moisture inversion methods holds significant importance for the high-quality development of the South-to-North Water Diversion Middle Route Project and for the regional economies involved.

Traditional soil moisture monitoring methods mainly include soil drying method, neutron moisture meter method, and tensiometer method, etc. They are all based on point measurement, which is high in accuracy but has a large workload, a long time period, and more demanding requirements, and cannot obtain data quickly. Conventional optical remote sensing (e.g., TM, SPOT, etc.), with fewer bands and low spectral resolution, is suitable for soil moisture monitoring over large areas, and has a large error for regional or plot-level soil moisture monitoring. In recent years, in order to obtain soil moisture in real-time, people have developed soil moisture sensors, but a single sensor can only realize the soil moisture monitoring of its buried location, and if you want to realize regional soil moisture monitoring, you need to bury a large number of sensors, which will cost a lot of manpower, material, and financial resources. In order to realize the soil moisture monitoring of the channel slopes in the deep excavated expansive soil section of the China South-to-North Water Diversion Project, through the on-site research, we found that in order to grasp the deformation status of the channel slopes in real-time, the project managers have installed a large number of GNSS observation stations in the risk areas of the channel slopes, and a huge amount of GNSS observation data can be acquired every day. At the same time, by reviewing the literature, we found that the monitoring of soil moisture can be realized by using GNSS-R technology. Based on the above, we fused the GNSS-R technique and deep learning method to carry out the soil moisture inversion study on the channel slope of the deep excavated square canal section of the South-to-North Water Diversion Central Route Project in China. GNSS-R technology utilizes the multipath reflection component of the signal-to-noise ratio (SNR) from GNSS satellites to retrieve near-surface physical parameters. In comparison with traditional soil moisture measurement methods, rapidly evolving remote sensing techniques offer numerous irreplaceable advantages for monitoring soil moisture. The use of GNSS-R involves the microwave frequency band (L-band), which exhibits strong penetration capabilities, reduced atmospheric attenuation, and excellent vegetation penetration. Consequently, it is considered an ideal frequency for soil moisture inversion at present [4]. Currently, GNSS-R technology has been widely extended to various application areas, including soil moisture [5], sea surface wind measurement [6], oil spill detection [7], and sea ice monitoring [8], among others.

Currently, European and American countries have conducted extensive fundamental research and experimentation on soil moisture detection techniques based on GNSS-R. In 2002, NASA included GPS dual-frequency radar measurements in the SMEX02 (Soil Moisture Experiment 2002) trial, which demonstrated the spatial and temporal correlations between the reflected signal strength and soil moisture [9]. In 2003, the UK-DMC satellite, which was equipped with GNSS-R instruments, successfully obtained physical parameters of the Earth's surface, such as sea surface roughness [10]. Additionally, high-precision elevation measurements can be derived from GPS reflection signals over calm sea areas [11]. During the period of 2013–2015, the Polytechnic University of Catalonia in

Spain conducted multiple ground-based and airborne GNSS-R experiments to carry out soil moisture measurements. These experiments involved both direct and reflected GNSS signal measurements while considering polarization. The researchers also took the instrument parameters that could affect the calculation of reflection coefficients into account [12–16]. With the emergence of unmanned aerial vehicles (UAVs) as a new remote sensing platform, there have been studies on soil moisture inversion with GNSS-R by using UAVs. In the field of direct and reflected signal interferometry with GNSS, Larson et al. proposed the GPS-MR (GPS–multipath reflectometry) soil moisture measurement technique. They utilized GPS signal-to-noise ratio (SNR) data from the Crustal Deformation Monitoring Network for this purpose [17]. Rodriguez-Alvarez et al. proposed the interference pattern technique (IPT) for soil moisture measurement. They used a custom-designed receiver and a vertically polarized antenna for this technique [18]. In China, research on GNSS-R technology started relatively later. In 2016, Han Moutian et al. derived a model for soil moisture inversion by using GNSS interferometric signal amplitudes based on the interference effect and the GNSS receiver signal-to-noise ratio estimation method. They also conducted a simulation verification of the proposed model [19]. In 2016, Yang Lei et al. and Zou Wenbo et al. conducted research on soil moisture measurement by using signals reflected from GEO satellites. They proposed empirical or analytical soil moisture inversion models based on their studies [20,21]. In 2018, Wu Jizhong et al. addressed the parameter estimation problem for obtaining soil moisture content by using GPS-IR (GPS–interferometric reflectometry). They proposed an improved method for estimating reflection signal parameters and studied the process of establishing soil moisture inversion models [22]. With the development of computer technology, scholars began to use deep learning techniques for soil moisture inversion. In 2019, Sun Bo et al. proposed a GA-SVM (genetic algorithm–support vector machine)-assisted method for soil moisture inversion and demonstrated through experiments that this method effectively improved the accuracy of soil moisture inversion [23]. In the same year, Zhang Nan et al. proposed a method for eliminating the micro-Doppler effect in GEO satellites for soil moisture inversion [24]. Zhang Xiaoyu et al. used GA-BP neural network to invert snow depth, effectively eliminating the jump phenomenon in the inversion process, reducing the error and improving the inversion accuracy [25]. In 2020, Zhu Chonghao et al. used the GABP neural network model to assess landslide risk in Sichuan Province as an example, and the results were better than BP neural network, which improved the efficiency of landslide risk assessment [26]. In 2021, Yang Lianbing et al. used a BP neural network optimized by a genetic algorithm to invert soil salinity, and the inversion results were better than the traditional BP neural network [27]. In the same year, Zhao Jianhui et al. used feature selection and GA-BP neural network to invert soil moisture, which provided a new idea for multi-source remote sensing surface soil moisture inversion in farmland [28]. In 2022, Schiajer performed soil moisture inversion using three neural networks, GABP, GRNN, and ELM, all of which achieved better inversion results [29]. In the same year, new mathematics study findings have been reported by researchers at Akdeniz University by comparing different ANN (Ffbp Grnn F) algorithms and multiple linear regression for daily streamflow prediction in Kocasu River, Turkey [30].

　　However, single-site single-satellite GNSS-R is unable to accurately monitor short-term variations in soil moisture, and the limited observation information obtained with a single satellite can result in significant differences in data quality [31]. To enhance the accuracy of the results, this study proposes a novel technique for multi-satellite and multi-frequency data fusion. This technique automatically selects satellites with a high correlation among the amplitude, phase, and soil moisture. It employs an adaptive fusion algorithm based on least squares to combine data from multiple satellites in the same frequency band. Furthermore, an entropy-based method is applied to fuse data from different frequency bands. By utilizing the fused data for GNSS-R technology, this study mitigates signal gaps and improves the quality of observational data, thereby enhancing the estimation accuracy of soil moisture. The data fusion approach helps fill in missing information and reduces the variability in the observations, leading to more reliable and precise estimations of soil

moisture. In order to validate the feasibility of the proposed method, this study employed the deep excavation of expansive soil channels in the South-to-North Water Diversion Project as the study area. By using deep learning techniques, models were established to correlate the phase, amplitude, and other relevant features with soil moisture. The results demonstrated that the soil moisture estimations obtained with the fused data from the proposed multi-satellite multi-frequency fusion technique outperformed those obtained from single-satellite single-frequency data inversion in terms of accuracy. This confirmed the effectiveness of the data fusion approach in improving the accuracy of soil moisture estimation when applied to the study area of the deep excavation of expansive soil channels in the South-to-North Water Diversion Project. In this paper, we use multi-satellite and multi-band data fusion techniques to process the GNSS-R observation data in order to obtain more comprehensive observation information, and use deep learning techniques to establish high-precision inversion models, which provide a new technical route for soil moisture inversion of deep excavated expansive soil channel slopes in the South-to-North Water Diversion Middle Route Project.

## 2. Basic Principles and Methods of GNSS-R Soil Moisture Inversion

### 2.1. GNSS-R Fundamentals

The GNSS-R reflectometry technique involves a dual-base radar that allows one to obtain surface roughness features and geophysical parameters, i.e., by using GNSS to measure the delay (time delay or phase delay) between the direct signal and the signal reflected from the surface mirror; then, based on the geometric positional relationships between GNSS satellites, receivers, and mirror reflection points, the surface features can be inverted [32]. When using geodetic receivers, the environmental noise level remains constant, so the signal-to-noise ratio directly corresponds to the strength of the GNSS signal that is received. Figure 1 depicts the direct and reflected signals received by the GNSS antenna, where the direct signal exhibits much higher intensity than that of the reflected signal. As shown in Figure 1, the interference between the direct signal and the reflected signal (or multipath signal) results in an overlay effect, causing oscillations, particularly at low satellite elevations. In most environments, the amplitude of the reflected signal is much smaller than that of the direct signal. Therefore, the signal-to-noise ratio is controlled by the direct signal, and the desired multipath effects can be extracted by separating this oscillation pattern.

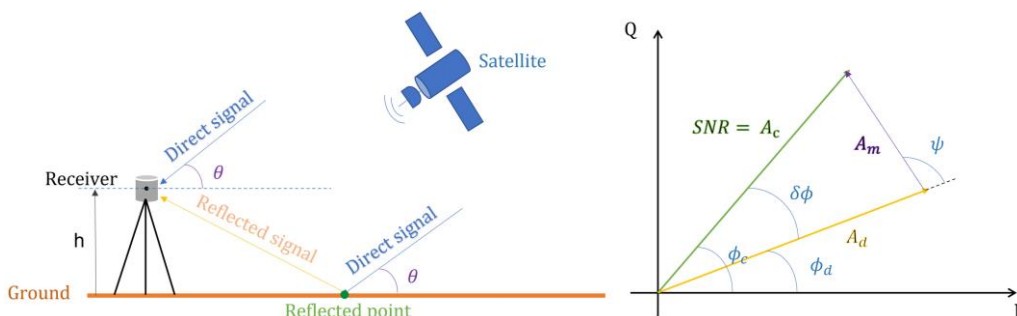

**Figure 1.** Diagram of the geometric relationships of the satellite, antenna, and reflector; diagram of the radiation relationships of the direct reflection signal.

The relationship between the *SNR* multipath amplitude and *SNR* is established by identifying the effect of the gain pattern of the receiving antenna on the recorded signal strength, and at any moment, the *SNR* and satellite altitude $\theta$ can be expressed by Equation (1) [33]:

$$SNR^2(\theta) = A_c^2(\theta) = A_d^2(\theta) + A_m^2(\theta) + 2A_d(\theta)A_m(\theta)cos\psi \tag{1}$$

where $A_d$ and $A_m$ represent the amplitudes of the direct signal and multipath signal, respectively, which indicate the contributions of the multipath signal to the SNR. $\psi$ denotes the phase difference between the two signals. $A_c$ is expressed as the composite signal amplitude of the two signals, i.e., the signal-to-noise ratio (SNR). Figure 2 shows the trend of SNR variation and altitude angle variation in the L1 band of the G02 satellite on 1 January 2021 at station GP01.

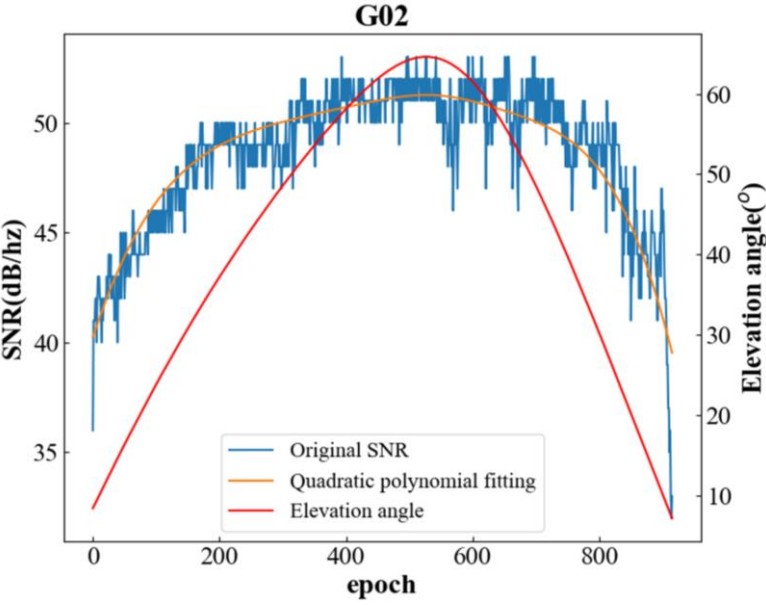

**Figure 2.** G02 satellite's SNR and the satellite's altitude angle.

As seen in Equation (1) and Figure 2, the change in the amplitude of the direct signal or multipath signal with respect to the phase leads to a corresponding change in the SNR amplitude, and the effect of the antenna gain pattern indicates that $A_d \gg A_m$. Thus, the overall amplitude of the SNR is mainly driven by the direct signal [34], while the multipath signal produces a small-amplitude, high-frequency oscillation in the direct signal and, thus, affects the SNR. This oscillation is more pronounced at lower satellite azimuth angles [35].

To determine the multipath amplitude of the SNR, it is necessary to separate the contribution of the multipath signal to the SNR from the amplitude of the direct signal $dSNR_{direct}$. This can be achieved by fitting a low-order polynomial to the SNR time series to estimate the direct signal and subtracting it from the original SNR data. The residual sequence, $dSNR_{multipath}$, represents the multipath component and can be expressed with Equation (2):

$$dSNR_{multipath} = SNR - dSNR_{direct} = A_m cos\left(\frac{4\pi h}{\lambda}sin\theta + \psi\right) \qquad (2)$$

In the equation, $A_m$ represents the amplitude, $\lambda$ denotes the carrier wavelength, $\psi$ represents the phase, and $h$ is the distance from the phase center of the receiving antenna to the reflecting surface, which is also known as the effective antenna height.

Due to the inability to obtain a complete periodic segment of the SNR's residual sequence in Figure 2, it is generally challenging to address it with a fast Fourier transform. However, the Lomb–Scargle algorithm can effectively extract weak periodic signals from non-uniform sequences [36]. Therefore, Lomb–Scargle spectral analysis is applied to the SNR's residual sequence to obtain the highest frequency, leading to the determination of the most effective vertical reflection height, h. During the fitting process for obtaining $\psi$ and $A_m$, the effective antenna height, h, is often treated as a fixed constant. However, in practical measurements, variations in satellite trajectories and environmental conditions around the receiver station can cause changes in h. In long-term observation sequences, the median of the effective antenna height is closest to the vertical distance from the receiver antenna

to the reflecting surface. Therefore, this study adopts the median value of the effective antenna height in the long-term observation sequence as a fixed value for h in the fitting of the feature parameters. The SNR reflection component exhibits periodic oscillations with the satellite elevation angle, approximating a cosine function. Therefore, a nonlinear least squares algorithm is employed to perform cosine fitting on the resampled data to obtain the reflection signal's amplitude parameter $A_m$ and phase parameter $\psi$. Finally, the soil moisture is inverted by using the amplitude parameter $A_m$ and phase parameter $\psi$.

### 2.2. Data Fusion Methods

This study proposes a novel data fusion method that utilizes multiple data processing algorithms for data preprocessing to enhance the inversion process. The method automatically selects satellites with a high correlation among the amplitude, phase, and soil moisture. An adaptive fusion algorithm based on least squares is then applied to merge data from multiple satellites in the same frequency band. Furthermore, an entropy-based fusion method is employed to merge amplitude and phase data from different frequency bands. By using this method, signal gaps are reduced, and the limitations of the limited observation information from a single satellite and varying data quality are addressed, resulting in improved data quality and enhanced accuracy in soil moisture inversion.

Before performing data fusion, to reduce the significant differences in amplitude caused by different satellites, the amplitude sequence is first arranged in ascending order. The average value of the top 20% of the sequence is selected as the baseline for normalization, as shown in Equation (3):

$$A_{norm} = \frac{A}{\overline{A_{20\%}}} \tag{3}$$

For the phase, the initial phase of each satellite signal arriving at the ground is different. In order to clearly derive the phase variation caused by humidity changes for the comparison of the phase characteristics of different satellites, etc., the phase time series of each satellite track needs to be zeroed, i.e., the minimum value is set to zero. When zeroing according to Equation (4), first, the average of the lowest 20% of observations for each track (satellite) is calculated, and then this average is subtracted from the phase time series.

$$\Delta\psi = \psi - \overline{\psi_{20\%}} \tag{4}$$

In the above equation, $\overline{A_{20\%}}$ represents the average of the top 20% of the largest values in the amplitude sequence, and $\overline{\psi_{20\%}}$ represents the average of the bottom 20% of the smallest values in the phase sequence. By applying the aforementioned processing, noise and errors caused by vegetation, terrain, and other factors can be removed from the time series, which is beneficial for soil moisture inversion. This step helps enhance the accuracy of soil moisture retrieval by mitigating the impacts of various sources of interference.

After normalizing the data, the same frequency band data from multiple satellites are fused by using an adaptive fusion algorithm based on the least squares method. The adaptive fusion algorithm based on the least squares method aims to minimize the total variance with respect to the true value by adjusting the weights of each datum, thus achieving more accurate fusion results. For the SNR observations provided by multiple satellites, the phase and amplitude values of each satellite are obtained after processing. These phase and amplitude data are then fused to obtain a more precise estimation. In this algorithm, the least squares method is used to solve for the optimal weighting coefficients that minimize the sum of squared errors between the fused result and the true value. Specifically, assuming that there are n satellites providing observations, and after processing, the phase data $x_1, x_2, \ldots, x_n$ is obtained, along with their corresponding weight coefficients $w_1, w_2, \ldots, w_n$, the objective is to solve the optimal weight coefficients that bring the

weighted result closest to the true value y. Then, the problem can be transformed into the following problem of minimizing an objective function:

$$min_{w_1, w_2, \dots w_n} \sum_{i=1}^{n} w_i (x_i - y)^2 \tag{5}$$

Equation (5) is derived so that the derivative is zero to solve for the optimal weight coefficients $w_1, w_2, \dots, w_n$. The optimal weighting factor can be expressed as Equation (6):

$$w = \left( X^T X \right)^{-1} X^T y \tag{6}$$

where $X = [x_1, x_2, \dots, x_n]$, y is the true value, and $w = [w_1, w_2, \dots, w_n]$ is the weight coefficient. Specifically, for the $i$th weighting factor,

$$w_i^* = \frac{1}{\sum_{j=1}^{n} \left( \frac{x_i - x_j}{x_i - y} \right)^2} \tag{7}$$

The optimal weighting coefficients are calculated according to Equation (7), and they are used to weigh the observations to obtain a more accurate estimate.

Finally, the entropy method is used to fuse data from different frequency bands acquired by GNSS receivers for fusion in order to obtain higher-quality observation data and improve the inversion accuracy. Data fusion with the entropy method involves multivariate data fusion based on the principle of information entropy. The core idea is that a greater information entropy indicates a greater uncertainty of the index and a smaller weight; a smaller information entropy indicates a lower uncertainty of the index and a larger weight. The observed values of each indicator are quantified according to certain rules, and then the information entropy and weight of each indicator are calculated; the final fusion result is calculated through the information entropy principle and the weighted average principle. Specifically, the entropy value method is calculated as follows:

(1) The normalized values of each column in the data are calculated and scaled to a range of [0, 1]; the formula is shown in Equation (8):

$$x_{ij} = \frac{x - x_{min}}{x_{max} - x_{min}} \tag{8}$$

In Equation (8), $x$ represents the original data, $x_{min}$ denotes the minimum value of the data, and $x_{max}$ corresponds to the maximum value of the data.

(2) The weight of the $i$th sample under the $j$th indicator is calculated for that indicator; the formula is shown in Equation (9):

$$p_{ij} = \frac{x_{ij}}{\sum_{i=1}^{m} x_{ij}} \tag{9}$$

(3) The entropy value is calculated for each column of data; the definition of entropy is used to compute the entropy value for each column of data. Entropy represents the uncertainty or information content of the data, and the formula for calculating the entropy value is shown in Equation (10):

$$E_j = -\sum_{i=1}^{m} p_{ij} \log p_{ij} \tag{10}$$

(4) The weights of each datum are calculated; the formula for calculating the weight $w_j$ for the $j$th indicator is shown in Equation (11):

$$w_j = \frac{1 - E_j}{\sum_{i=1}^{m} (1 - E_k)} \tag{11}$$

(5)    The formula for performing data fusion is shown in Equation (12):

$$x_i = \sum_{j=1}^{n} w_j p_{ij} \tag{12}$$

The fusion algorithm automatically selects satellites with high correlation between amplitude–phase data and soil moisture, fuses data from multiple satellites in multiple segments, reduces signal loss, improves the quality of observation data, and obtains high-precision inversion models.

### 2.3. Soil Moisture Inversion Based on Deep Learning

#### 2.3.1. BP Neural Network

The artificial neural network algorithm, as the name suggests, is an algorithmic network composed of artificial neurons that mimics the way neural transmission occurs in the human brain. It possesses strong capabilities for nonlinear mapping, self-organization, adaptation, memory, and prediction, making it well suited for solving complex logical operations and nonlinear problems [37]. Neural networks can be used for tasks such as classification, clustering, and prediction. They require a sufficient amount of historical data, and by training on this data, a network can learn the underlying knowledge within the data. The BP neural network, which is a widely used and a classical artificial neural network, possesses the aforementioned capabilities, along with characteristics such as strong plasticity, simplicity, and powerful learning abilities. Today, it is used in extensive applications across various fields [38].

The BP neural network is the fundamental form of a neural network, and its output is obtained through forward propagation, while the error is propagated back through the network by using a backpropagation method. A BP neural network emulates the activation and propagation processes of human neurons. Considering a three-layer neural network as an example, a BP neural network consists of three layers: the input layer, the hidden layer, and the output layer. The input layer receives data, and the output layer outputs data. Each neuron in the previous layer is connected to neurons in the next layer, collecting information from the previous layer and transmitting it to the next layer through activation. The structure of a BP neural network is depicted in Figure 3.

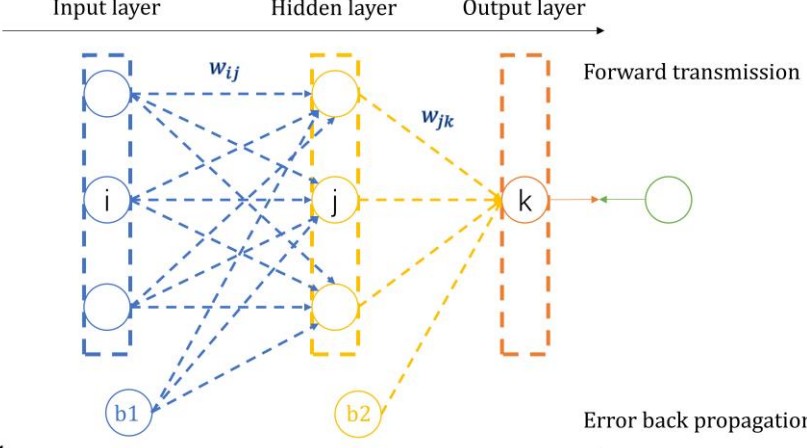

**Figure 3.** Diagram of the structure of a BP neural network. The blue arrow represents the input layer to the hidden layer, while the brown arrow represents the hidden layer to the output layer.

Here, i is the number of input layer neurons, j is the number of hidden layer neurons, k is the number of output layer neurons, w is the weight, and b is the "bias". Each circle is a neuron.

The BP algorithm includes the following two processes: (1) Forward propagation of information, where the feature signal is passed forward along the input layer and passed

to the output layer nodes through the hidden layer's neurons. The output nodes do not directly output the signal but need to undergo a series of nonlinear changes. The obtained output signal is analyzed for error with the target output signal, and if the error is too large, it is transferred to the error backpropagation process. (2) In the backward propagation of error, the error obtained from the forward propagation of the signal is reversed from the output layer to the entire neural network; the error is divided equally among the nodes in each layer when it passes through the hidden layer and the input layer, and the network weights are updated so that the error decreases layer by layer along the reversed neural network and so on until the forward propagation of the signal reaches the desired output. The threshold and weights corresponding to the actual output are determined at this time, and the training of the neural network can be stopped. Specifically, assuming a three-layer BP neural network with M input layer nodes, N hidden layer nodes, and O output layer nodes and by using a sigmoid function as the activation function, the main steps of BP neural network training are as follows.

(1) The input variables $net_i$ are computed for the $i$th node of the hidden layer of the neural network; the equation is shown in Equation (13):

$$net_i = \sum_{j=1}^{M} w_{ij} x_i + \theta_i \qquad (13)$$

where the meaning of the variable $x_i$ denotes the input parameter of the $j$th node of the input layer, $j = 1, \ldots, M$; the meaning of the variable $w_{ij}$ denotes the neural network's weight parameter between the $i$th node of the hidden layer and the $j$th node of the input layer; the meaning of the variable $\theta_i$ denotes the threshold parameter of the ith node of the hidden layer.

(2) The output variable $y_i$ is computed for the $i$th node of the hidden layer of the neural network; the equation is shown in Equation (14):

$$y_i = g(net_i) = g\left( \sum_{j=1}^{M} w_{ij} x_i + \theta_i \right) \qquad (14)$$

where $g(x)$ is the excitation function of the hidden layer. A sigmoid function expressed by Equation (15) is used in this study:

$$g(x) = \frac{1}{1 + e^{-x}} \qquad (15)$$

(3) The input variable $net_k$ is calculated for the $k$th node of the output layer of the neural network; the equation is shown in Equation (16):

$$net_k = \sum_{i=1}^{q} w_{ki} y_i + a_k = \sum_{i=1}^{q} w_{ki} g\left( \sum_{j=1}^{M} w_{ij} x_i + \theta_i \right) + a_k \qquad (16)$$

where the meaning of the variable $w_{ki}$ denotes the weight parameter between the $k$th node of the output layer and the $i$th node of the hidden layer, $i = 1, \ldots, q$; the variable $a_k$ denotes the threshold parameter of the $k$th node of the output layer, $k = 1, \ldots, L$;

(4) The output variable $o_k$ is computed for the $k$th node of the output layer of the neural network; the equation is shown in Equation (17):

$$o_k = g(net_k) = g\left( \sum_{i=1}^{q} w_{ki} y_i + a_k \right) = g\left( \sum_{i=1}^{q} w_{ki} g\left( \sum_{j=1}^{M} w_{ij} x_i + \theta_i \right) + a_k \right) \qquad (17)$$

(5)  The error $E$ is calculated with Equation (18):

$$E = \frac{1}{2} \sum_{k=1}^{M} (Y_k - o_k) \tag{18}$$

where $Y_k$ is the desired output.

(6)  The weight is updated with Equation (19):

$$\begin{cases} w_{ij} = w_{ij} + \eta y_i(1 - y_i)x_i \\ w_{jk} = w_{jk} + \eta y_i(Y_k - o_k) \end{cases} \tag{19}$$

where $\eta$ is the learning rate.

(7)  The threshold is updated with Equation (20):

$$\begin{cases} \theta_i = \theta_i + \eta y_i(1 - y_i) \\ a_k = a_k + \eta(Y_k - o_k) \end{cases} \tag{20}$$

(8)  It is determined whether the iteration of the algorithm is finished, and if not, one returns to step (2).

A flowchart of the BP neural network is shown in Figure 4.

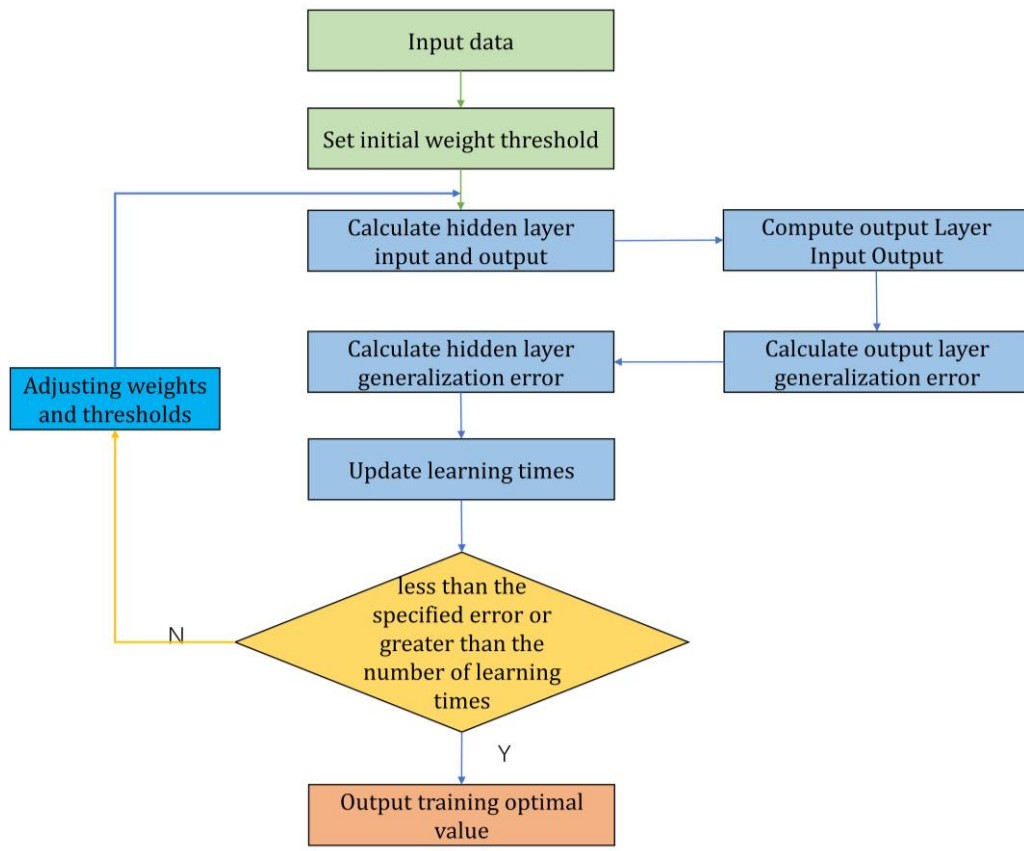

**Figure 4.** Flowchart of BP neural network training.

### 2.3.2. GA-BP Neural Network

While BP neural networks have strong learning capabilities and robustness, they can suffer from some limitations. Because their search mechanism is that of gradient descent, without prior knowledge, the initial values and weights of the network are random, making them prone to being trapped in local minima instead of finding the global minimum. Consequently, a network may fail to obtain the optimal solution, and its learning and

memory can be unstable. If training samples are added, the pre-trained network needs to be retrained from the beginning without leveraging the previous knowledge of weights and thresholds. This increases the learning burden and reduces the learning efficiency [39]. To address these issues, the genetic algorithm (GA) can be used to optimize a BP neural network. By incorporating the GA, it is possible to quickly obtain the optimal neural network parameters, accelerate the learning process, and enhance the progress of network inversion.

Genetic algorithms are used to construct a fitness function based on the objective function of a problem, evaluate and perform genetic operations, select a population consisting of multiple solutions (each solution corresponds to a chromosome), and reproduce it over multiple generations to obtain the individual with the best fitness value as the optimal solution to the problem [40]. The specific steps are as follows:

(1) Chromosome encoding: A real-number encoding strategy is used to implement the encoding of the chromosomes of the genetic algorithm. The S-order real matrix is set to $[-1, 1]$, based on which the parameters, such as the connection weights between nodes in each layer of the BP neural network and node thresholds in the hidden layer and output layer, are encoded and solved for optimality [41]. Compared with binary coding, real-number coding does not require decoding at a later stage, the coding length is shorter, and the accuracy of the parameter search is high [42].

(2) Initializing the population: The initial population of $W = (w_1; w_2; \ldots; w_p)$ is randomly generated, and the number of individuals in the population is set to P. Individuals $w_i$, $w_1; w_2; \ldots; w_s$ are generated with a linear interpolation function for one chromosome of the algorithm.

(3) Calculation of the population individuals' fitness values: The sum of the squared training errors is used for the calculation of the population individuals' fitness values.

(4) Selection: By using the roulette wheel method, the selection probability can be calculated with Equation (21):

$$p_i = \frac{f_i}{\sum_{i=1}^{p} f_i} \tag{21}$$

where $f_i$ is the fitness function, and p is the population size.

(5) Crossover: The crossover operation of gene $w_q$ at position j and the crossover operation of gene $w_s$ at position j are performed according to Equation (22):

$$\begin{cases} w_{qj} = w_{qj}(1-b) + w_{sj}b \\ w_{sj} = w_{sj}(1-b) + w_{qj}b \end{cases} \tag{22}$$

where $b$ is a random number in the range of [0,1].

(6) Mutation: The $j$th gene of the $i$th individual undergoes population variation, and the operation of which can be described by Equations (23) and (24):

$$w_{ij} = \begin{cases} w_{ij} + (w_{ij} - w_{max})f(g) & r \geq 0.5 \\ w_{ij} + (w_{min} - w_{ij})f(g) & r < 0.5 \end{cases} \tag{23}$$

$$f(g) = r_2\left(1 - \frac{g}{G_{max}}\right) \tag{24}$$

where $w_{max}$ and $w_{min}$ are the maximum and minimum values of gene $w_{ij}$, respectively; $G_{max}$ is the maximum number of evolutions; $g$ is the current iteration number; $r$ is a random number in the range of [0, 1]; $r_2$ is a random number.

(7) Obtaining new populations: Steps (4) to (6) are repeated until the optimal solution is output.

A flowchart of GA-BP neural network training is shown in Figure 5.

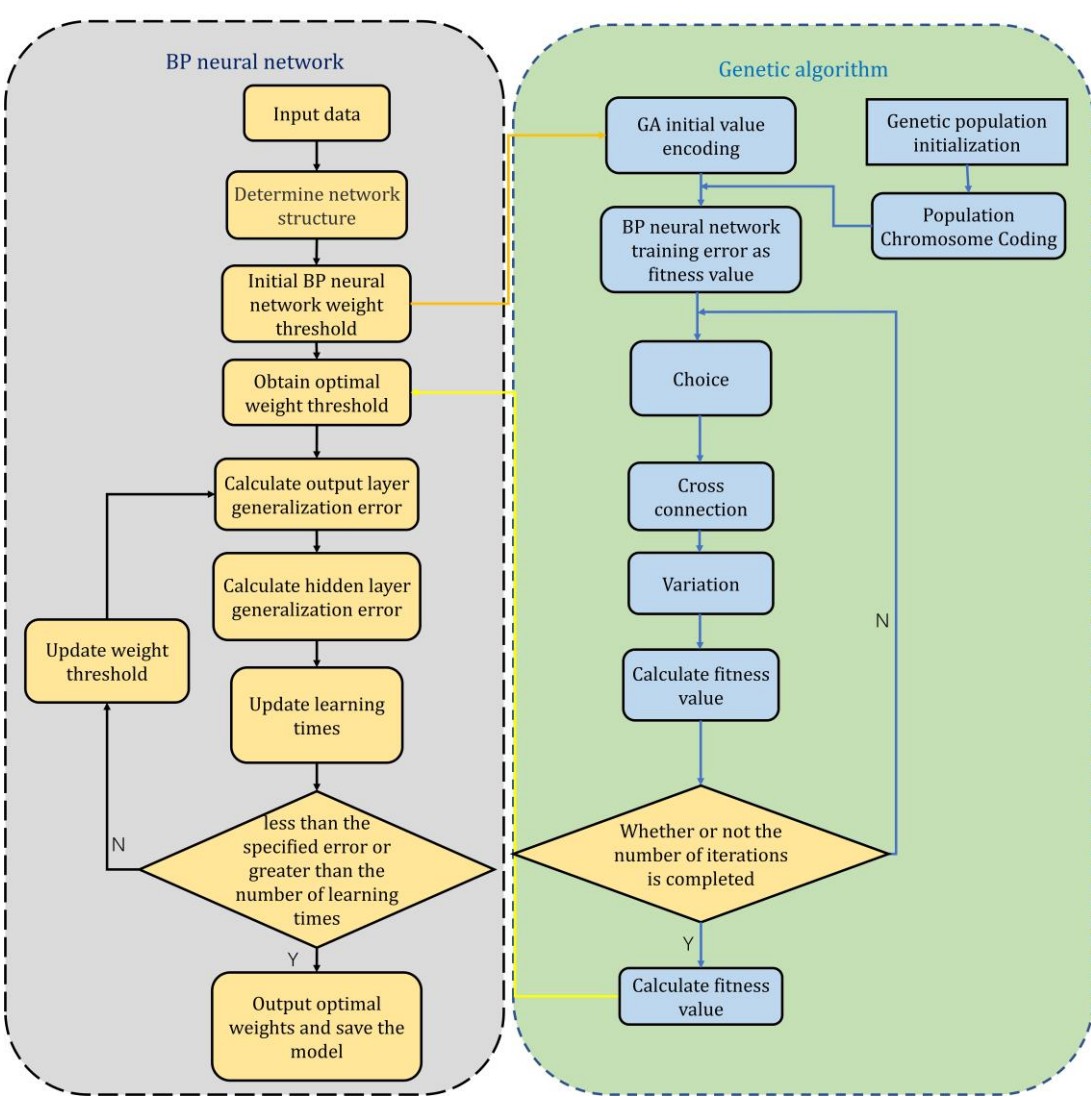

**Figure 5.** Flowchart of GA-BP neural network training.

### 3. Experimental Area and Data Sources

*3.1. Experimental Area*

The study area was the head canal section of the South–North Water Diversion Project in China, which was located at the head of the Tao Fork Canal in Danyang Village, Jiu Chong Town, Xi Chuan County, Nanyang City, Henan Province, terminating at the junction of Fangcheng County and Ye County, with the end pile number 185 + 545 and a total length of 185.545 km, of which the channel length was 176.718 km and the building length was 8.827 km. Here, there were 58.411 km of deep excavation canals, with a maximum depth of 47.5 m and an opening width of 373.22 m; there were 33.689 km of fill canals, with a maximum fill height of 17 m; there were 149.476 km of swelling soil canals, accounting for 84.5% of the total canal length and including 56.729 km of weak swelling soil canals, 84.37 km of medium swelling soil canals, and 8.377 km of strong swelling soil canals. The experimental area is shown in Figure 6.

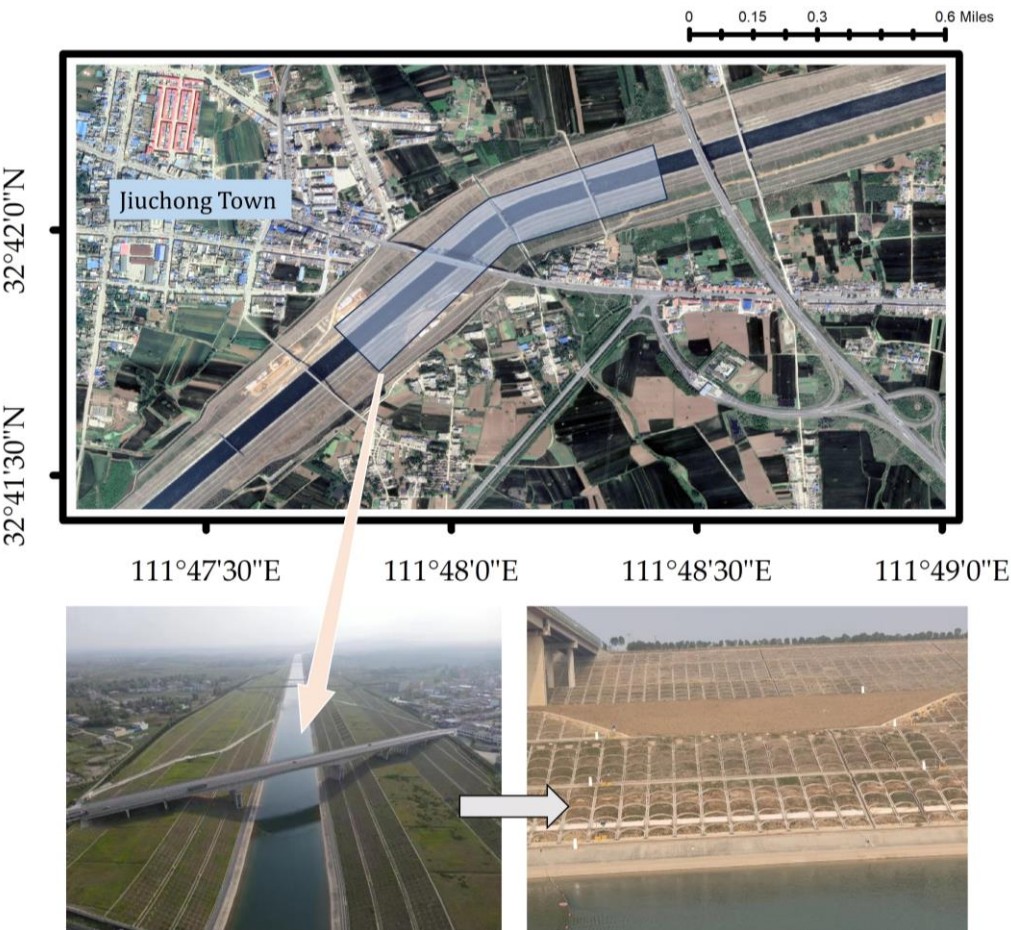

**Figure 6.** Real map of the GNSS study area.

*3.2. Experimental Data*

The data adopts multi-system data from three GNSS automated measurement stations (GP01, GP02, GP03) in the experimental area, and uses continuous observation satellite data from three stations for a total of 150 days from 16 December 2020 to 14 May 2021 to conduct a multi-system combination GNSS-R high-resolution soil moisture inversion study. A soil moisture sensor probe was buried at a depth of approximately 7.5 cm about 1 m next to the GNSS receiver device, and the soil moisture communication device and GNSS receiving device were integrated. To validate the performance of the method, in situ soil moisture data measured near the station sites were used as reference data for comparison. The modeling data consisted of 125 days of satellite data and the corresponding soil moisture priors collected from 16 December 2020 to 19 April 2021. The data for verifying the model accuracy, on the other hand, comprised satellite data and soil moisture priors collected from 20 April 2021 to 14 May 2021. All GNSS receiver data can be used, and the Beiyun receiver used in this study. The GNSS station map and soil moisture meter map is shown in Figure 7, the GNSS monitoring receiver is shown in Figure 8, and the parameter configuration is shown in Table 1.

**Table 1.** GNSS receiver parameter configuration.

| Receiver Performance Indicators | |
| --- | --- |
| tracking channel | GPS L1C/A, L2C, L2P |
| positioning accuracy | 3 h positioning accuracy: plane: ±1–3 mm, elevation: ±2–5 mm. |
| data sampling rate | Supports up to 50 Hz |
| data logging | 32 GB TF card; supporting data storage and transmission; Supports circular storage. |
| timing accuracy | 20 ns RMS |
| Speed measurement accuracy | 0.03 m/s RMS |
| power supply | 12 V DC input; the power is less than 2 W. |

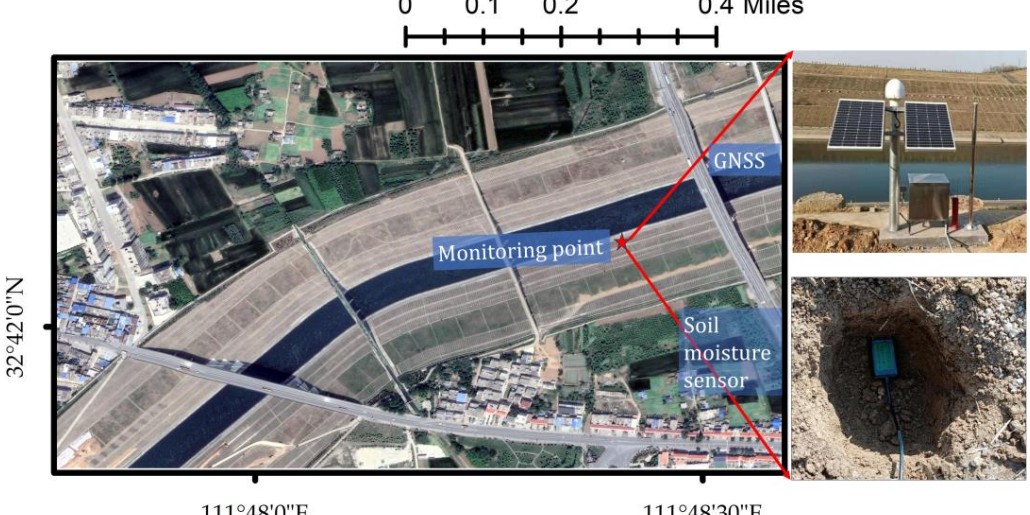

**Figure 7.** Real map of the GNSS stations and soil hygrometer.

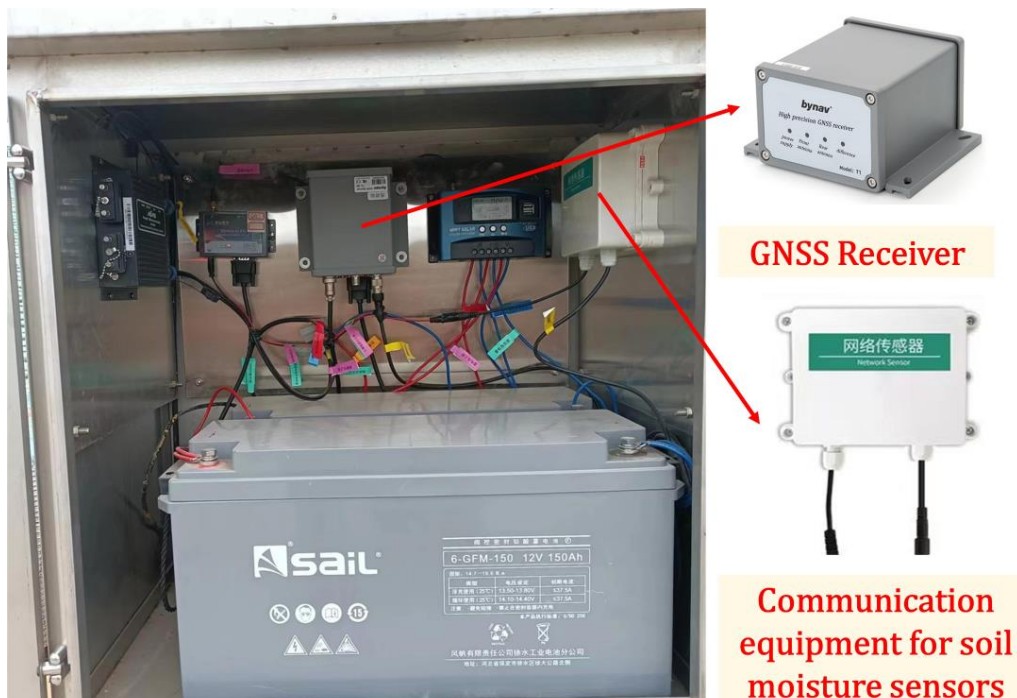

**Figure 8.** GNSS receiver diagram and Soil moisture sensors communication equipment.

## 4. Experiments and Results

### 4.1. Experimental Technical Program

Figure 9 shows a flowchart of the soil moisture inversion technique used in this study. As can be seen in the figure, the technical route of this study can be divided into three steps: (1) preprocessing of the GNSS-R soil moisture inversion data to extract the characteristic parameters of the amplitude and phase of the reflected signal from the observation data acquired with the original GNSS receiver; (2) the use of the multi-satellite multi-frequency data fusion technique to fuse the acquired characteristic parameter data to obtain more accurate observation data and improve the inversion accuracy; and (3) a linear model, BP neural network model, and GA-BP neural network model are developed to invert the soil moisture and compare the inversion accuracy of single-satellite data with that of the fused data.

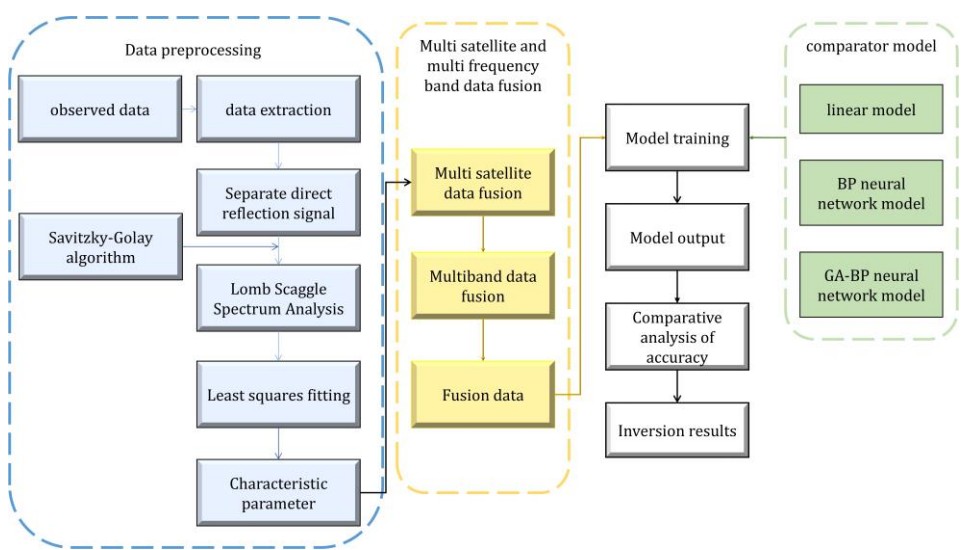

**Figure 9.** Flowchart of the soil moisture inversion technology.

### 4.2. Extraction of the Reflected Signal's Feature Parameters

The GNSS receiver's observation data are collected in the carrier phase and pseudor-ange format, while GNSS-R soil moisture retrieval requires the utilization of the satellite elevation angle and signal-to-noise ratio (SNR). To obtain these parameters, it is necessary to calculate them from the GNSS observation and navigation files with the relevant equations.

In this study, the signal-to-noise ratio data were extracted from the navigation file with the satellite altitude angle data by using teqc software. At low satellite altitude angles, the signal-to-noise ratio had a serious multipath effect and showed periodic oscillations. With the gradual increase in the satellite altitude angle, the antenna had a larger gain, and the signal-to-noise ratio tended to be stable. In order to extract the reflected signal data from the GNSS SNR data, a low-order polynomial fit to the SNR data was used to separate the contribution of the multipath effect to the SNR from the amplitude of the direct signal and to remove the direct component. In addition, in order to standardize the SNR data in dB/Hz units, which are normally converted into values in linear units (Volts/Volts), the linearization formula shown in Equation (25) was used [43]:

$$SNR_{V/V} = 10^{SNR_{dB/Hz}/20} \tag{25}$$

Figure 10a shows a plot of the signal-to-noise ratio versus the satellite altitude angle. The signal-to-noise ratio data are shown in blue, and the direct radiation signal data from the low-order polynomial fit are shown in red. Figure 10b shows the linearized reflected signal after removing the direct signal.

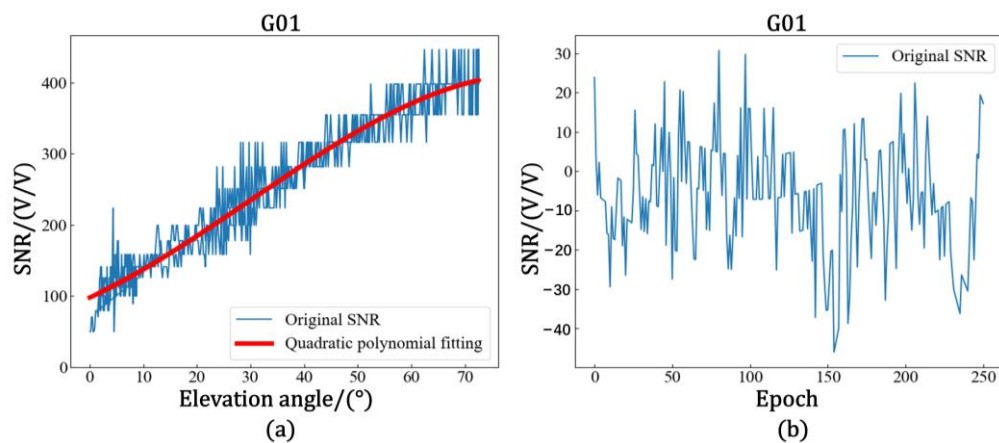

**Figure 10.** (**a**) SNR data and satellite altitude angle map. (**b**) Linearized reflection signal graph after removing the direct signal.

The research area is a slope with a gentle slope, with a maximum slope of around 15 degrees, which affects signal reception. In order to eliminate the influence of surface environmental factors, the Savitzky–Golay algorithm was introduced to preprocess the multipath components [44] in order to remove the effects of noise and coarse differences. Figure 11 shows the reflected signal after processing with the SG algorithm.

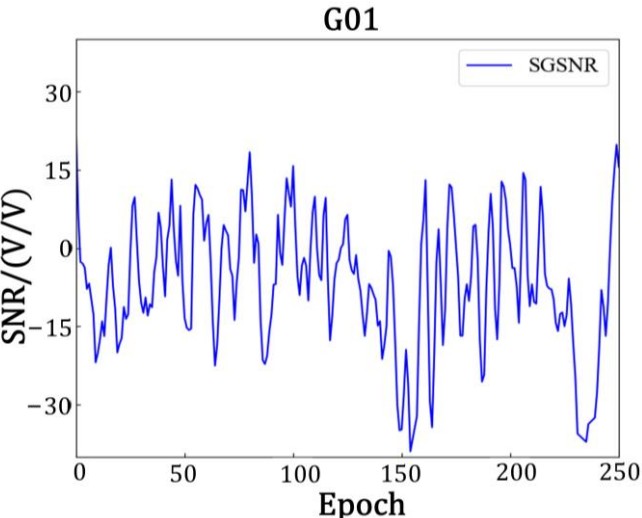

**Figure 11.** Reflected signal processed with the Savitzky–Golay algorithm.

Lomb–Scargle spectrum analysis (LSP) was used to estimate the temporal variations in the principal frequency of the signal-to-noise ratio interferogram to obtain the principal frequency; this was obtained as the effective reflector height h according to Equation (26). Figure 12 shows the LSP spectrum analysis of station GP02 and satellite G01 on 1 January 2021.

$$h = \frac{1}{2}\lambda f \tag{26}$$

where $f$ is the main frequency, $\lambda$ is the carrier wavelength, the wavelength of the L1 band is 24.42 cm, and that of the L2 band is 19.03 cm.

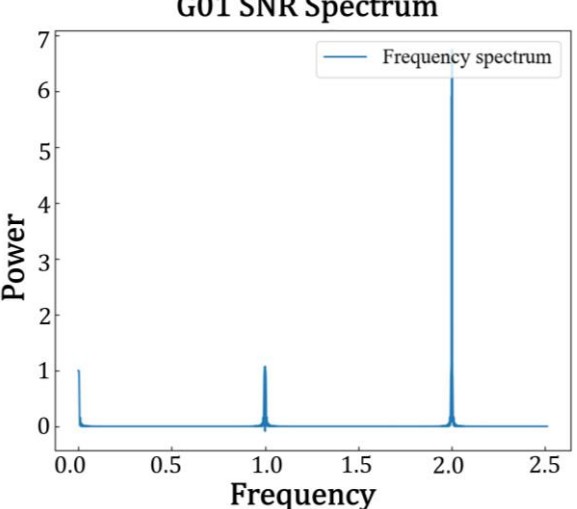

**Figure 12.** LSP spectrum analysis of station GP02 and satellite G01 on 1 January 2021.

After obtaining the h values, the characteristic parameters of the amplitude and delayed phase were obtained by fitting the nonlinear least squares in Equation (2) [45]. Table 2 shows some of the amplitude and phase data obtained by fitting station GP02.

**Table 2.** Phase and amplitude data of station GP02.

| Satellite | Maximum Elevation/° | Minimum Elevation/° | Frequency Band | Amplitude/v | Phase/rad |
|---|---|---|---|---|---|
| G29 | 24.93 | 2.47 | L1 | 3.987 | 3.053 |
| G05 | 24.94 | 7.34 | L1 | 5.570 | 4.126 |
| G19 | 24.99 | 2.07 | L1 | 4.911 | 3.883 |
| G26 | 24.95 | 4.06 | L1 | 7.780 | 2.324 |
| G22 | 24.97 | 2.79 | L2 | 3.485 | 3.884 |
| G14 | 24.95 | 3.34 | L2 | 6.362 | 1.788 |
| G32 | 24.98 | 2.90 | L2 | 5.067 | 2.916 |
| G21 | 24.92 | 2.02 | L2 | 2.180 | 3.308 |

*4.3. Data Fusion*

Before data fusion, in order to reduce the excessive differentiation of the feature elements caused by different satellites, the inverse-derived data were first normalized by using Equations (3) and (4) for the amplitude and phase data, respectively. Table 3 shows the normalized data of some characteristic elements of station GP02. Figure 13 shows the normalized feature elements of some satellites of station GP02.

**Table 3.** Normalization phase and amplitude data of station GP02.

| Satellite | Maximum Elevation/° | Minimum Elevation/° | Frequency Band | Amplitude/v | Phase/rad |
|---|---|---|---|---|---|
| G29 | 24.93 | 2.47 | L1 | 0.818 | 1.446 |
| G05 | 24.94 | 7.34 | L1 | 0.502 | 2.190 |
| G19 | 24.99 | 2.07 | L1 | 0.766 | 1.533 |
| G26 | 24.95 | 4.06 | L1 | 0.991 | 1.310 |
| G22 | 24.97 | 2.79 | L2 | 0.489 | 2.982 |
| G14 | 24.95 | 3.34 | L2 | 0.445 | 1.140 |
| G32 | 24.98 | 2.90 | L2 | 0.385 | 2.246 |
| G21 | 24.92 | 2.02 | L2 | 0.178 | 1.787 |

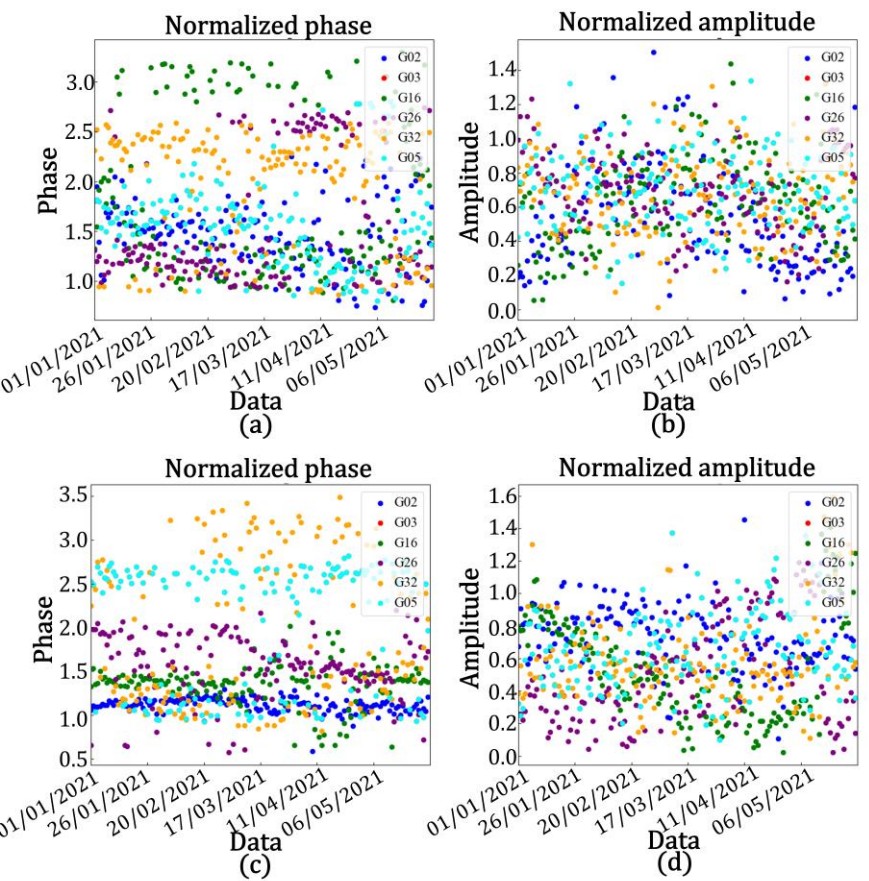

**Figure 13.** (**a**) Phase data map of the L1 frequency band of station GP02 after the normalization of multiple satellites. (**b**) Amplitude data map of the L1 frequency band of station GP02 after the normalization of multiple satellites. (**c**) Phase data map of the L2 frequency band of station GP02 after the normalization of multiple satellites. (**d**) Amplitude data map of the L1 frequency band of station GP02 after the normalization of multiple satellites.

The normalized characteristic elements were subjected to a correlation analysis with soil moisture. The results of the correlation analysis are shown in Table 4.

**Table 4.** Correlation analysis between characteristic elements and soil moisture.

| Monitoring Station | Feature | Frequency Band | Correlation Coefficient ($R^2$) | | | | |
|---|---|---|---|---|---|---|---|
| | | satellite | G06 | G19 | G22 | G29 | G05 |
| GP01 | Phase | L1 | −0.586 | −0.566 | −0.493 | −0.492 | −0.474 |
| | | L2 | −0.610 | −0.574 | −0.487 | −0.414 | 0.356 |
| | Amp | L1 | −0.470 | −0.391 | −0.340 | 0.312 | 0.301 |
| | | L2 | 0.582 | 0.564 | −0.523 | −0.494 | 0.486 |
| GP02 | Phase | L1 | −0.375 | −0.535 | −0.426 | 0.505 | 0.275 |
| | | L2 | −0.451 | 0.440 | 0.578 | −0.532 | 0.503 |
| | Amp | L1 | 0.335 | 0.422 | −0.302 | 0.292 | −0.287 |
| | | L2 | −0.370 | 0.451 | 0.469 | 0.308 | −0.243 |
| GP03 | Phase | L1 | −0.776 | −0.668 | 0.549 | −0.485 | 0.485 |
| | | L2 | −0.715 | 0.672 | 0.427 | 0.394 | 0.393 |
| | Amp | L1 | −0.469 | 0.464 | −0.436 | −0.431 | 0.410 |
| | | L2 | −0.705 | 0.683 | −0.649 | −0.607 | 0.606 |

According to the correlation results in Table 4, it can be seen that for points GP01 and GP03, the phase and amplitude of the L1 and L2 bands of satellite G06 had a strong correlation with soil moisture; for point GP02, the phase and amplitude of satellite G19 had

a strong correlation with soil moisture in the L1 band, while the phase and amplitude of satellite G22 had a strong correlation with soil moisture in the L2 band. Therefore, in this study, satellite G06 was selected for subsequent single-satellite inversion experiments for points GP01 and GP03, satellite G19 was selected for subsequent single-satellite inversion experiments for point GP02 in the L1 band, and satellite G22 was selected for subsequent single-satellite inversion experiments for point GP02 in the L2 band. At the same time, the four satellites with the highest correlations were automatically selected by using the fusion algorithm for the joint multi-satellite inversion experiments with the corresponding satellites at the corresponding points. Table 5 shows the fusion of the single-satellite data with multi-satellite multi-band data on 16 December 2020 for the data on the characteristic parameters.

**Table 5.** Table of the characteristic parameters for 16 December 2020.

| Monitoring Station | Satellite | Maximum Elevation/° | Minimum Elevation/° | Frequency Band | Amplitude/v | Phase/rad |
|---|---|---|---|---|---|---|
| GP01 | G06 | 24.98 | 6.32 | L1 | 1.061 | 1.098 |
| | G06 | 24.98 | 6.32 | L2 | 0.901 | 1.157 |
| | | 24.98 | 2.37 | Fusion | 0.370 | 0.277 |
| GP02 | G19 | 24.99 | 2.07 | L1 | 0.850 | 1.754 |
| | G22 | 24.97 | 2.79 | L2 | 0.617 | 1.007 |
| | | 24.99 | 2.07 | L2 | 0.280 | 0.277 |
| GP03 | G06 | 24.97 | 2.10 | L1 | 0.559 | 1.752 |
| | G06 | 24.97 | 2.10 | L2 | 1.012 | 1.241 |
| | | 24.98 | 2.02 | Fusion | 0.271 | 0.279 |

*4.4. Soil Moisture Inversion Results*

In order to verify the effectiveness of the fusion algorithm, and considering that the deep learning algorithm had self-learning and adaptive abilities for solving high-dimensional nonlinear problems, models of three methods were established for a comparative analysis. The three models included the conventional linear regression model, the BP neural network model, and the GA-BP neural network model. We trained using the data from the first 125 days (for linear inversion experiments, we used 125 days of data to establish the model; for neural network experiments, we divide the data into training and validation sets in an 8:2 ratio to establish the model) and then used the 25 day data that did not participate in the training as the test set to verify the accuracy of the trained model, in order to verify its generalization ability and real performance. The linear model was modeled separately for the amplitude and phase with soil moisture, and it has been verified by many studies that the inversion effect of unifying the amplitude and phase as x-value inputs when using a deep learning algorithm is more accurate than when using single-feature element inversion [45]. Therefore, in this study, the amplitude, phase, and frequency were used as input x-values for the deep learning network, and soil moisture values were used as output y-values for training.

Figure 14 shows the linear model that was built at site GP01. Figure 15 shows the linear model that was built at site GP02. Figure 16 shows the linear model that was built at site GP03. Table 6 shows the results of the root mean square error (RMSE), model goodness of fit (R2), correlation (r), mean squared error (MSE), and mean absolute error (MAE) for the linear model's training.

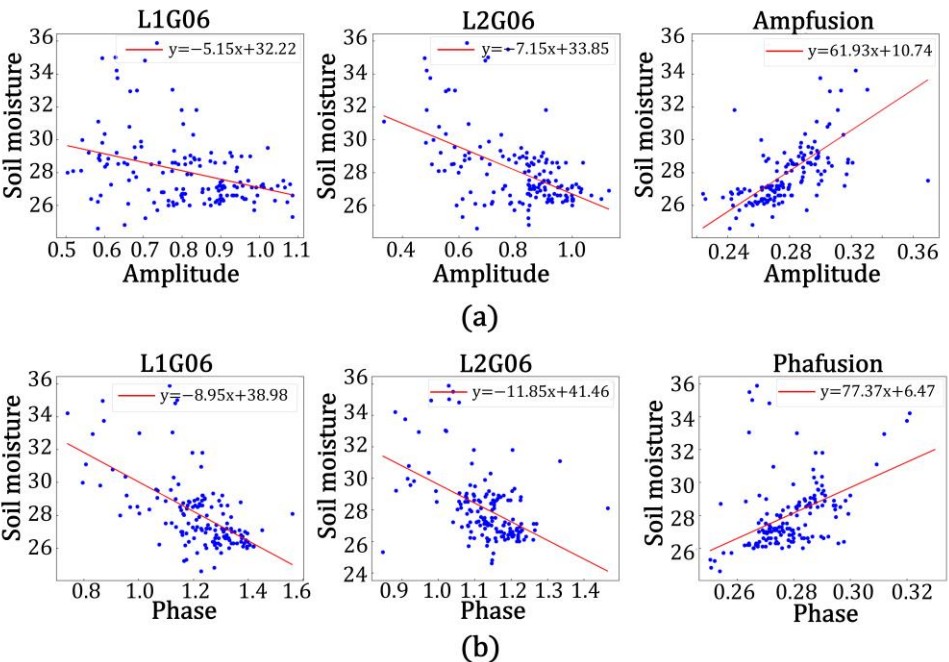

**Figure 14.** Linear model established for station GP01: (**a**) amplitude linear model; (**b**) phase linear model.

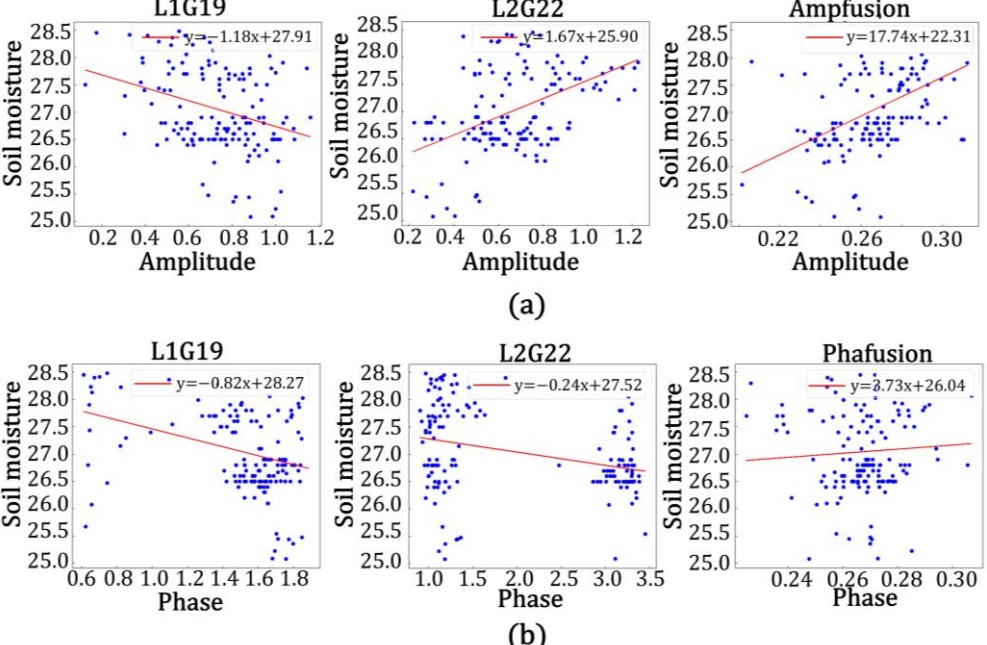

**Figure 15.** Linear model established for station GP02: (**a**) amplitude linear model; (**b**) phase linear model.

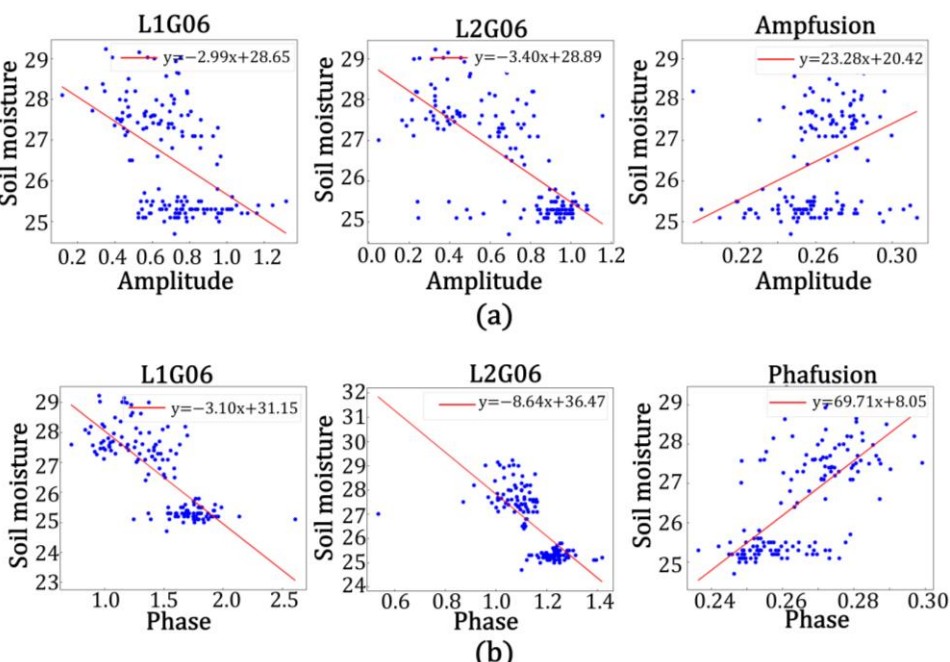

**Figure 16.** Linear model established for station GP03: (**a**) amplitude linear model; (**b**) phase linear model.

**Table 6.** Analysis of the accuracy of the linear model.

| Monitoring Station | Data | | RMSE | $R^2$ | r | MAE | MSE |
|---|---|---|---|---|---|---|---|
| | | L1 | 2.035 | 0.116 | 0.340 | 1.415 | 4.141 |
| | Amp | L2 | 1.844 | 0.274 | 0.523 | 1.290 | 3.400 |
| GP01 | | Fusion | 1.724 | 0.321 | 0.567 | 1.128 | 2.972 |
| | | L1 | 1.754 | 0.343 | 0.586 | 1.183 | 3.077 |
| | Phase | L2 | 1.890 | 0.237 | 0.487 | 1.339 | 3.572 |
| | | Fusion | 1.612 | 0.432 | 0.657 | 1.037 | 2.599 |
| | | L1 | 0.743 | 0.091 | 0.302 | 0.618 | 0.552 |
| | Amp | L2 | 0.688 | 0.218 | 0.467 | 0.559 | 0.473 |
| GP02 | | Fusion | 0.689 | 0.220 | 0.469 | 0.547 | 0.475 |
| | | L1 | 0.737 | 0.106 | 0.326 | 0.594 | 0.543 |
| | Phase | L2 | 0.740 | 0.097 | 0.311 | 0.587 | 0.548 |
| | | Fusion | 0.717 | 0.133 | 0.365 | 0.560 | 0.514 |
| | | L1 | 1.150 | 0.220 | 0.469 | 0.973 | 1.323 |
| | Amp | L2 | 1.211 | 0.135 | 0.368 | 1.055 | 1.467 |
| GP03 | | Fusion | 0.923 | 0.497 | 0.705 | 0.686 | 0.852 |
| | | L1 | 0.822 | 0.602 | 0.776 | 0.643 | 0.676 |
| | Phase | L2 | 0.911 | 0.511 | 0.715 | 0.698 | 0.830 |
| | | Fusion | 0.723 | 0.616 | 0.785 | 0.616 | 0.523 |

As shown in Figures 14–16 and Table 6, the linear model that was developed did reflect the relationship between the characteristic elements and soil moisture. The highest single-satellite inversion correlation for station GP01 was the phase inversion of the L1 band, with a correlation of 58.6%, RMSE of 1.754, MAE of 1.183, MSE of 3.077, and R2 of 0.343, and the highest single-satellite inversion correlation for station GP02 was the amplitude inversion of the L2 band, with a correlation of 46.7%, RMSE of 0.688, MAE of 0.559, MSE of 0.473, and R2 of 0.218. The highest single-satellite inversion correlation of station GP03 was the phase inversion of the L2 band, with a correlation of 71.5%, RMSE of 0.911, MAE of 0.698, MSE of 0.830, and R2 of 0.511; the highest correlation of the multi-satellite multi-band fusion data from station GP01 was in the phase inversion, with a correlation of 65.7%, RMSE of 1.724, MAE of 1.037, MSE of 2.599, and R2 of 0.432, while the highest correlation of the

multi-satellite multi-band fusion data from station GP02 was in the amplitude inversion, with a correlation of 46.9%, RMSE of 0.689, MAE of 0.547, MSE of 0.475, and R2 of 0.220. The highest correlation of the multi-satellite multi-band fusion data from station GP03 was in the phase inversion, with a correlation of 78.5%, RMSE of 0.723, MAE of 0.616, MSE of 0.523, and R2 of 0.616. The results in Table 6 show that the fused data had an improved correlation and model fit, and the root mean square error and mean absolute error decreased, which verified the effectiveness of the multi-satellite multi-band fusion technique with a linear model.

In this experiment, the training process of the BP neural network was set to 6000 epochs. One epoch is the process of importing the entire dataset for complete training, with 16 hidden layers and a learning rate of 0.02. In the GA algorithm of GABP neural network, the population size is set to 500, the mutation rate is 0.09, the crossover rate is 0.1, and the number of iterations is 500; in the BP neural network, the number of epochs is set to 1000, and the learning rate is 0.01. Figure 17 shows the loss value curve of the training set and validation set during the training process of BP neural network, and Figure 18 shows the loss value curve of the training set and validation set during the training process of GABP neural network. As shown in the Figure, there is no overfitting during the training process.

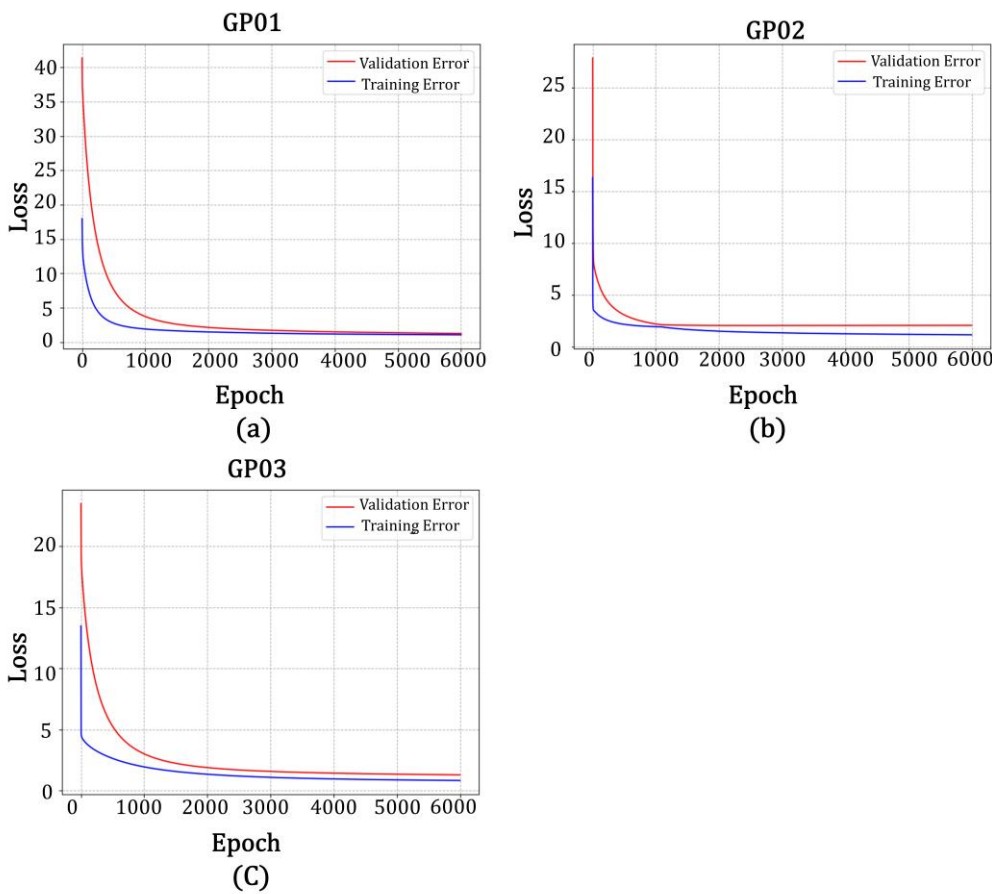

**Figure 17.** BP neural network training process of the training set and validation set of the loss value curve (**a**) GP01 station training process loss value curve. (**b**) GP02 station training process loss value curve. (**c**) GP03 station training process loss value curve.

After the training, we used untrained data from 20 April 2021 to 14 May 2021 as test data to verify the accuracy of the model. Figure 19 shows the results of soil moisture values in the linear model test set. Figure 20 shows the soil moisture results of the BP neural network test set. Figure 21 shows the soil moisture results of the GA-BP neural network model test set.

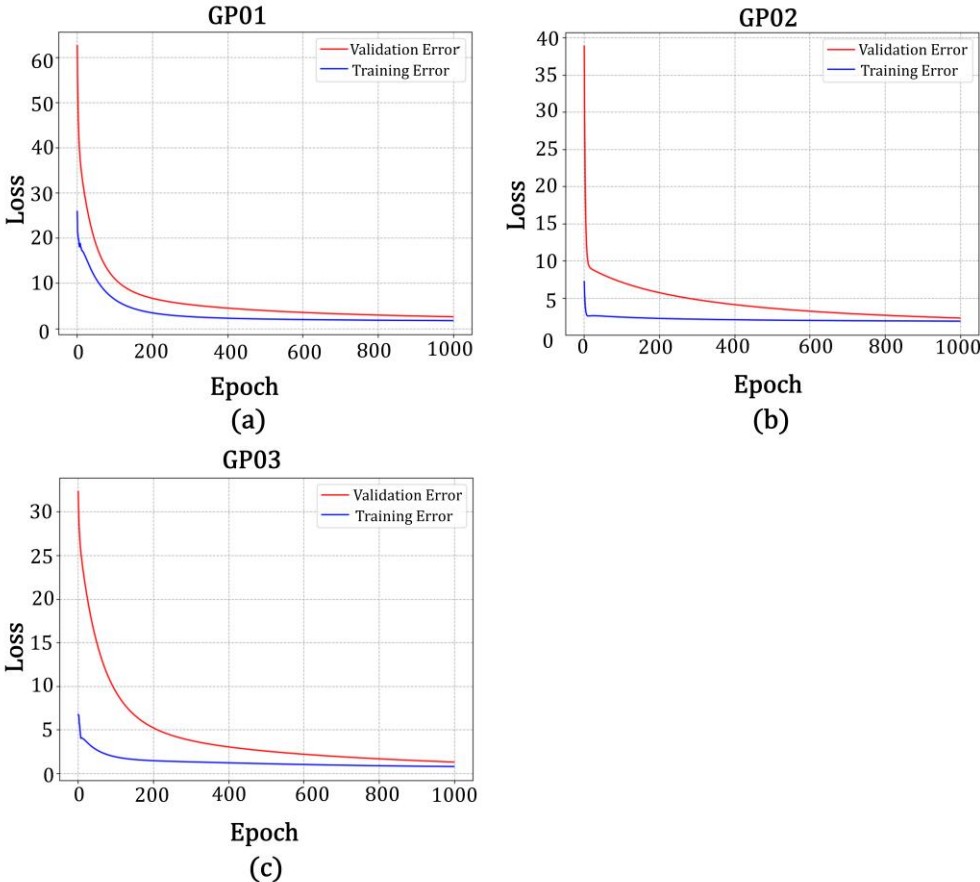

**Figure 18.** GABP neural network training process of the training set and validation set of the loss value curve (**a**) GP01 station training process loss value curve. (**b**) GP02 station training process loss value curve. (**c**) GP03 station training process loss value curve.

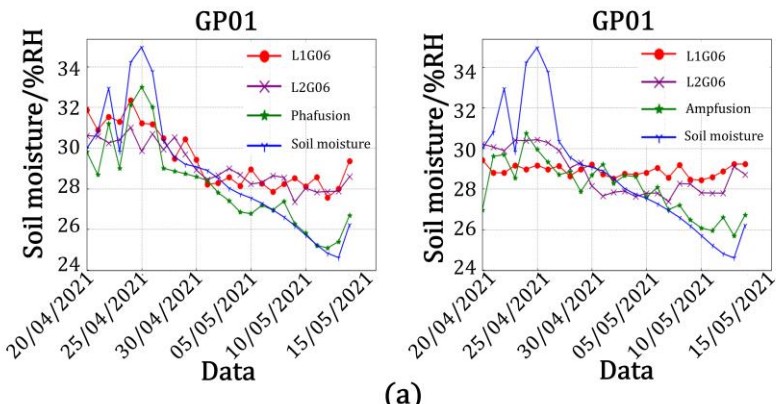

(a)

**Figure 19.** *Cont.*

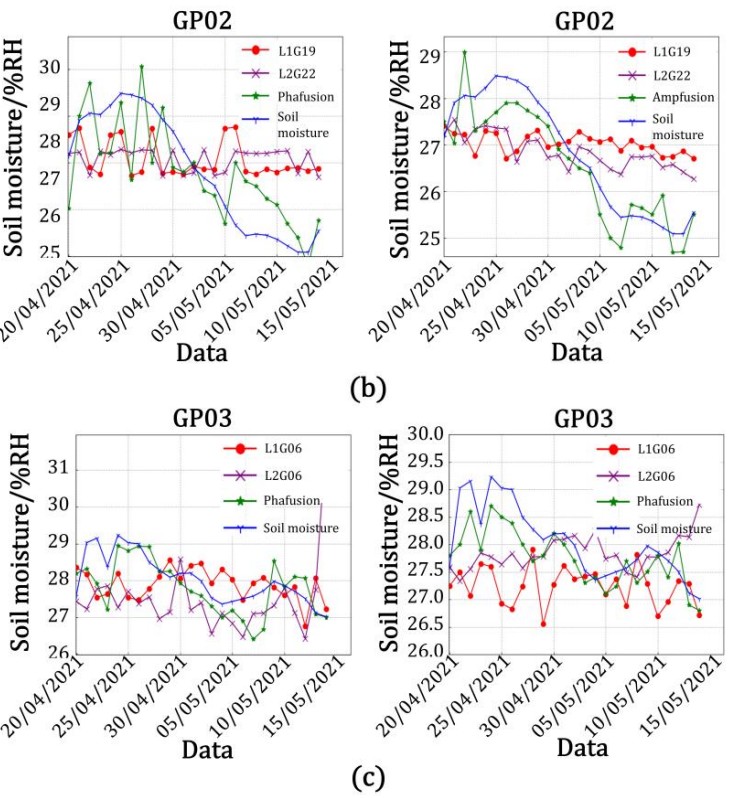

**Figure 19.** The linear model's prediction results for soil moisture. (**a**) Frequency band and fusion data for predicting soil moisture values for station GP01. (**b**) Frequency band and fusion data for predicting soil moisture values for station GP02. (**c**) Frequency band and fusion data for predicting soil moisture values for station GP03.

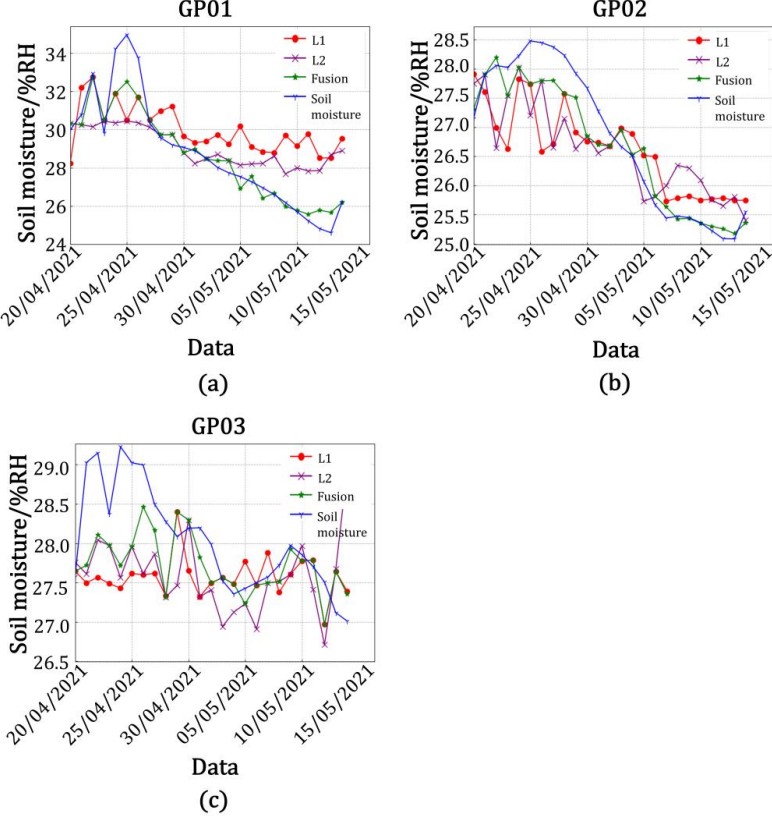

**Figure 20.** BP neural network model for predicting soil moisture. (**a**) Frequency band and fusion data

for predicting soil moisture values for station GP01. (**b**) Frequency band and fusion data for predicting soil moisture values for station GP02. (**c**) Frequency band and fusion data for predicting soil moisture values for station GP03.

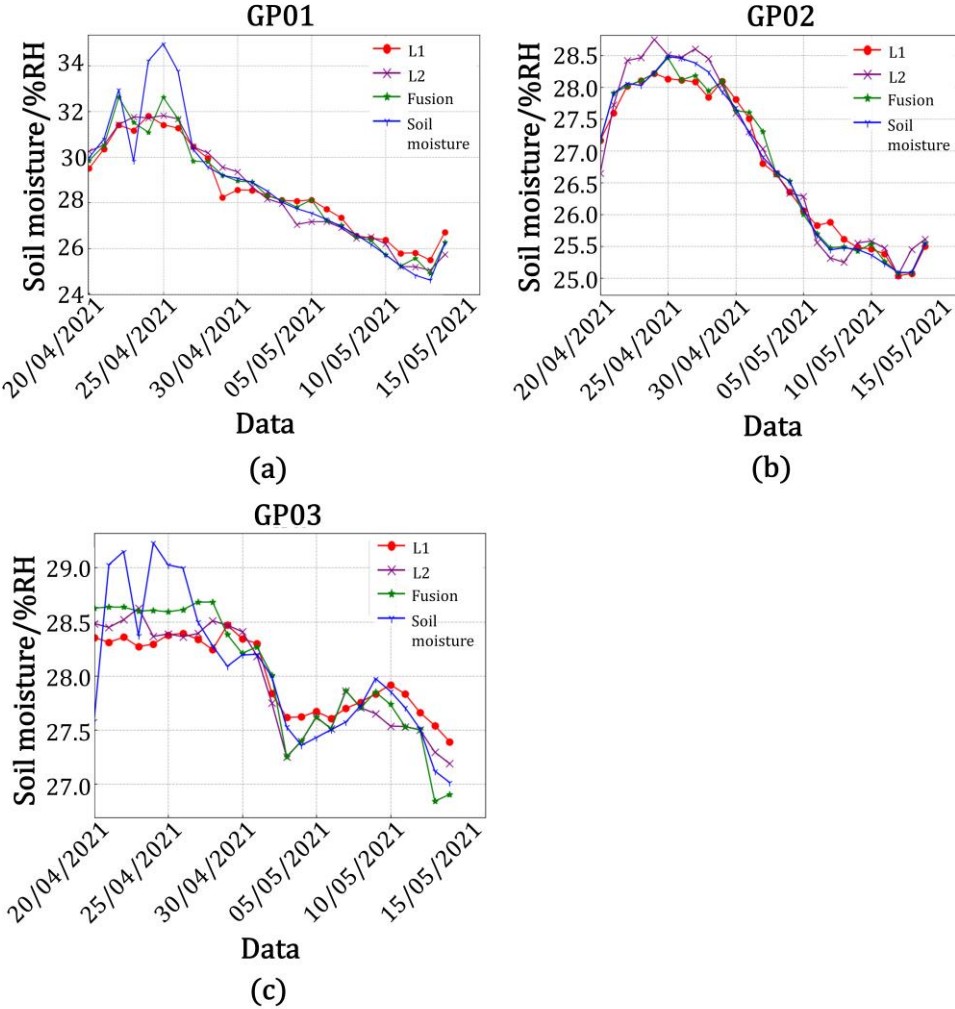

**Figure 21.** GA-BP neural network model for predicting soil moisture. (**a**) Frequency band and fusion data for predicting soil moisture values for station GP01. (**b**) Frequency band and fusion data for predicting soil moisture values for station GP02. (**c**) Frequency band and fusion data for predicting soil moisture values for station GP03.

## 5. Discussion

In this study, five metrics—the root mean square error (*RMSE*), model goodness of fit (*R2*), correlation (*r*), mean absolute error (*MAE*): and mean squared error (*MSE*)—were used to evaluate the models' accuracy.

Root mean square error (*RMSE*): The square root of the ratio of the square of the deviation of the observed value from the true value to the number of observations *N*. This reflects the extent to which the measured data deviate from the true value. The formula is shown in Equation (27):

$$RMSE = \sqrt{\frac{1}{N}\sum_{i=1}^{N}(y_i - Y_i)^2} \tag{27}$$

where $y_i$ is the predicted value of soil moisture, and $Y_i$ is the true value of soil moisture.

Model goodness of fit ($R^2$): The percentage of the variance in the dependent variable y that can be explained by the independent variable x. The formula is shown in Equation (28)–(31):

$$\overline{Y} = \frac{1}{N} \sum_{i=1}^{N} Y_i \tag{28}$$

$$SS_{tot} = \sum_{i=1}^{N} (Y_i - \overline{Y})^2 \tag{29}$$

$$SS_{res} = \sum_{i=1}^{N} (y_i - Y_i)^2 \tag{30}$$

$$R^2 = 1 - \frac{SS_{res}}{SS_{tot}} \tag{31}$$

Correlation ($r$): This measures the correlation between the predicted and actual values. The formula is shown in Equation (32):

$$r = \frac{COV(y, Y)}{\sqrt{VAR(y)VAR(Y)}} = \frac{\sum (y_i - \overline{y})(Y - \overline{Y})}{\sqrt{\sum (y_i - \overline{y}) \sum (Y - \overline{Y})}} \tag{32}$$

Mean absolute error ($MAE$): This is the average of the absolute error between the predicted value and the actual value. The formula is shown in Equation (33):

$$MAE = \frac{\sum_{i=1}^{N} |y_i - Y_i|}{N} \tag{33}$$

Mean Squared Error ($MSE$): This is a commonly used measure of the difference between the predicted values of a model and the actual observed values to assess how well the model fits on the given data. The formula is shown in Equation (34):

$$MSE = \frac{1}{N} \sum_{i=1}^{N} (y_i - Y_i)^2 \tag{34}$$

Table 7 shows the results for the root mean square error ($RMSE$), model goodness of fit ($R^2$), correlation ($r$), mean absolute error ($MAE$), and mean squared error ($MSE$) between the predicted and true values of soil moisture for the linear model.

As shown in Figure 19 and Table 7, the accuracy of the model built by using the data from the L2 band was higher than that when using the L1 band in the model of amplitude and soil moisture at station GP01, with the correlation between the predicted and true values of L2 band being 76.0%, the root mean square error being 2.144, the goodness of fit being 0.578, the mean square error being 4.597, and the mean absolute error being 1.608. The correlation between the predicted and true values of the fused data was 87.7%, the root mean square error was 1.927, the goodness of fit was 0.769, the mean square error being 3.713, and the mean absolute error was 1.365. It was calculated that the correlation between the predicted and true values of the fused data improved by 12.7%, the root mean square error decreased by 0.217, the goodness of fit improved by 0.191, the mean square error decreased by 0.884, and the mean absolute error decreased by 0.243 in comparison with the single-satellite data on the L2 band. In the model, the accuracy of the model built with data from the L2 band was higher than that built with data from the L1 band. The correlation between the predicted and true values of the L2 band was 88.9%, the root mean square error was 1.921, the goodness of fit was 0.790, the mean square error being 3.690, the mean absolute error was 1.560, and the correlation between the predicted and true values of fused data was 98.4%; the root mean square error was 1.028, the goodness of fit was 0.968, the mean square error being 1.057, and the mean absolute error was 0.790. It was calculated that the correlation between the predicted and true values of the fused data was

improved by 9.5%, the root mean square error was reduced by 0.893, the goodness of fit was improved by 0.178, the mean square error decreased by 2.633, and the mean absolute error was reduced by 0.770 in comparison with the single-satellite data in the L2 band. Similarly, it could be calculated that the correlation between the predicted and true values of the fused data in the amplitude and soil moisture model for station GP02 was improved by 7.6%, the root mean square error was reduced by 0.476, the fit was improved by 0.251, the mean square error decreased by 0.719, and the mean absolute error was reduced by 0.449 in comparison with the single-satellite data for the L2 band. In the phase and soil moisture model for station GP02, the correlation between the predicted and true values of the fused data improved by 7.6%, the root mean square error decreased by 0.472, the fit improved by 0.127, the mean square error decreased by 0.969, and the mean absolute error decreased by 0.458 in comparison with the single-satellite data for the L2 band. In the amplitude and soil moisture model for station GP03, the correlation between the predicted and true values of the fused data improved by 9.3%, the root mean square error decreased by 0.636, the fit improved by 0.151, the mean square error decreased by 0.940, and the mean absolute error decreased by 0.463 in comparison with the single-satellite data for the L1 frequency band. In the phase and soil moisture model for station GP03, the correlation between the predicted and true values of the fused data improved by 6.5%, the root mean square error decreased by 0.18, the fit improved by 0.087, the mean square error decreased by 0.240, and the mean absolute error decreased by 0.162 in comparison with the single-satellite data for the L1 band.

**Table 7.** Analysis of the accuracy of the linear model between the predicted and true values of soil moisture.

| Monitoring Station | Data | | RMSE | $R^2$ | r | MAE | MSE |
|---|---|---|---|---|---|---|---|
| GP01 | Amp | L1 | 2.792 | 0.558 | 0.747 | 2.195 | 7.795 |
| | | L2 | 2.144 | 0.578 | 0.760 | 1.608 | 4.597 |
| | | Fusion | 1.927 | 0.769 | 0.877 | 1.365 | 3.713 |
| | Phase | L1 | 2.521 | 0.716 | 0.846 | 1.356 | 6.355 |
| | | L2 | 1.921 | 0.790 | 0.889 | 1.560 | 3.690 |
| | | Fusion | 1.028 | 0.968 | 0.984 | 0.790 | 1.057 |
| GP02 | Amp | L1 | 1.192 | 0.616 | 0.785 | 1.089 | 1.421 |
| | | L2 | 0.993 | 0.731 | 0.855 | 0.903 | 0.986 |
| | | Fusion | 0.517 | 0.867 | 0.931 | 0.454 | 0.267 |
| | Phase | L1 | 1.233 | 0.537 | 0.733 | 1.087 | 1.520 |
| | | L2 | 1.263 | 0.632 | 0.795 | 1.115 | 1.595 |
| | | Fusion | 0.791 | 0.759 | 0.871 | 0.657 | 0.626 |
| GP03 | Amp | L1 | 1.057 | 0.623 | 0.789 | 0.822 | 1.117 |
| | | L2 | 0.850 | 0.627 | −0.792 | 0.644 | 0.723 |
| | | Fusion | 0.421 | 0.778 | 0.882 | 0.359 | 0.177 |
| | Phase | L1 | 0.760 | 0.411 | 0.641 | 0.614 | 0.578 |
| | | L2 | 1.377 | 0.392 | −0.626 | 1.029 | 1.896 |
| | | Fusion | 0.581 | 0.498 | 0.706 | 0.452 | 0.338 |

Table 8 shows the results for the root mean square error (*RMSE*), model goodness of fit ($R^2$), correlation (*r*), mean absolute error (*MAE*), and mean squared error (*MSE*) between the predicted and true values of soil moisture from the BP neural network model.

As shown in Figure 20 and Table 8, the accuracy of the model built by using the data from the L2 band in the BP neural network model of station GP01 was higher than the accuracy of that built with the data from the L1 band. The correlation between the predicted and true values for the L2 band was 84.2%, the root mean square error was 2.114, the goodness of fit was 0.447, the mean square error was 4.469, and the mean absolute error was 1.615. The correlation between the predicted and true values for the fused data was 96.4%, the root mean square error was 0.907, the goodness of fit was 0.898, the mean

square error was 0.823, and the mean absolute error was 0.602. It was calculated that the correlation between the predicted and true values of the fused data was improved by 12.2%, the root mean square error was reduced by 1.207, the goodness of fit was improved by 0.122, the mean square error was decreased by 3.646, and the mean absolute error was reduced by 1.013 in comparison with the single-satellite data in the L2 band. In the BP neural network model for station GP02, the accuracy of the model built with the data from the L2 band was higher than that of the model built with data from the L1 band. The correlation between the predicted and true values for the L2 band was 81.1%, the root mean square error was 0.787, the goodness of fit was 0.594, the mean square error was 0.619, and the mean absolute error was 0.651. The correlation between the predicted and true values for the fused data was 96.5%, the root mean square error was 0.392, the goodness of fit was 0.899, the mean square error was 0.154, and the mean absolute error was 0.298. It was calculated that the correlation between the predicted and true values of the fused data was improved by 15.4%, the root mean square error was reduced by 0.395, the goodness of fit was improved by 0.305, the mean square error was decreased by 0.465, and the mean absolute error was reduced by 0.353 in comparison with the single-satellite data for the L2 band. In the BP neural network model for station GP03, the accuracy of the model built with the data from the L1 band was higher than that of the model built with data from the L2 band. The correlation between the predicted and true values of the L1 band was 70.1%, the root mean square error was 0.826, the goodness of fit was 0.671, the mean square error was 0.682, and the mean absolute error was 0.635. The correlation between the predicted and true values for the fused data was 75.9%, the root mean square error was 0.599, the goodness of fit was 0.720, the mean square error was 0.359, and the mean absolute error was 0.432. Compared with the single-satellite data for the L1 band, the correlation between the predicted and true values of the fused data was improved by 5.8%, the root mean square error was reduced by 0.227, the goodness of fit was improved by 0.058, the mean square error was decreased by 0.323, and the mean absolute error was reduced by 0.203.

**Table 8.** Analysis of the accuracy of the BP neural network model between the predicted and true values of soil moisture.

| Monitoring Station | Data | RMSE | $R^2$ | r | MAE | MSE |
|---|---|---|---|---|---|---|
| GP01 | L1 | 2.459 | 0.252 | 0.734 | 2.102 | 6.047 |
| | L2 | 2.114 | 0.447 | 0.842 | 1.615 | 4.469 |
| | Fusion | 0.907 | 0.898 | 0.964 | 0.602 | 0.823 |
| GP02 | L1 | 0.805 | 0.575 | 0.807 | 0.679 | 0.648 |
| | L2 | 0.787 | 0.594 | 0.811 | 0.651 | 0.619 |
| | Fusion | 0.392 | 0.899 | 0.965 | 0.298 | 0.154 |
| GP03 | L1 | 0.826 | 0.671 | −0.701 | 0.635 | 0.682 |
| | L2 | 0.843 | 0.642 | 0.684 | 0.677 | 0.711 |
| | Fusion | 0.599 | 0.720 | 0.759 | 0.432 | 0.359 |

Table 9 shows the results for the root mean square error (*RMSE*), model goodness of fit ($R^2$), correlation (*r*), mean absolute error (*MAE*), and mean squared error (*MSE*) between the predicted and true values of soil moisture for the GA-BP neural network model.

As shown in Figure 21 and Table 9, the accuracy of the model built with the data from the L2 band in the GA-BP neural network model for station GP01 was higher than that of the model built with data from the L1 band. The correlation between the predicted and true values for the L2 band was 89.1%, the root mean square error was 1.078, the goodness of fit was 0.856, the mean square error was 1.162, and the mean absolute error was 0.688, and the correlation between the predicted and true values of the fused data was 95.4%. The correlation between the predicted and true values of the fused data was 95.4%, the root mean square error was 0.983, the goodness of fit was 0.880, the mean square error was 0.966, and the mean absolute error was 0.533. It was calculated that the correlation between

the predicted and true values of the fused data improved by 6.3%, the root mean square error decreased by 1.207, the goodness of fit improved by 0.024, the mean square error was decreased by 0.196, and the mean absolute error decreased by 0.155 in comparison with the single-satellite data for the L2 band. In the GA-BP neural network model for station GP02, the accuracy of the model built with the data from the L2 band was higher than that of the model built with the data from the L1 band. The correlation between the predicted and true values for the L2 band was 88.5%, the root mean square error was 0.199, the goodness of fit was 0.874, the mean square error was 0.040, and the mean absolute error was 0.151. The correlation between the predicted and true values of the fused data was 94.2%, the root mean square error was 0.154, the goodness of fit was 0.885, the mean square error was 0.024, and the mean absolute error was 0.096. It was calculated that the correlation between the predicted and true values of the fused data was improved by 5.3%, the root mean square error was reduced by 0.045, the goodness of fit was improved by 0.011, the mean square error was decreased by 0.016, and the mean absolute error was reduced by 0.055 in comparison with the single-satellite data for the L2 band. In the GA-BP neural network model for station GP03, the accuracy of the model built with the data from the L1 band was higher than that of the model built with the data from the L2 band. The correlation between the predicted and true values for the L1 band was 82.2%, the root mean square error was 0.409, the goodness of fit was 0.590, the mean square error was 0.167, and the mean absolute error was 0.308. The correlation between the predicted and true values of the fused data was 84.8%, the root mean square error was 0.342, the goodness of fit was 0.713, the mean square error was 0.117, and the mean absolute error was 0.250. It was calculated that the correlation between the predicted and true values of the fused data was improved by 2.6%, the root mean square error was reduced by 0.067, the goodness of fit was improved by 0.123, the mean square error was decreased by 0.050, and the mean absolute error was reduced by 0.058 in comparison with the single-satellite data for the L1 band.

**Table 9.** Analysis of the accuracy of the GA-BP neural network model between the predicted and true values of soil moisture.

| Monitoring Station | Data | RMSE | $R^2$ | r | MAE | MSE |
|---|---|---|---|---|---|---|
| | L1 | 1.173 | 0.830 | 0.852 | 0.820 | 1.376 |
| GP01 | L2 | 1.078 | 0.856 | 0.891 | 0.688 | 1.162 |
| | Fusion | 0.983 | 0.880 | 0.954 | 0.533 | 0.966 |
| | L1 | 0.238 | 0.863 | 0.885 | 0.190 | 0.057 |
| GP02 | L2 | 0.199 | 0.874 | 0.889 | 0.151 | 0.040 |
| | Fusion | 0.154 | 0.885 | 0.942 | 0.096 | 0.024 |
| | L1 | 0.409 | 0.590 | 0.822 | 0.308 | 0.167 |
| GP03 | L2 | 0.399 | 0.609 | 0.791 | 0.308 | 0.159 |
| | Fusion | 0.342 | 0.713 | 0.848 | 0.250 | 0.117 |

We analyzed the soil moisture inversion error, calculated the absolute soil moisture inversion error (the difference between the inversion error and the true value), and analyzed the interval distribution pattern.

Table 10 shows the maximum, median, and minimum true values of soil moisture, the predicted values of soil moisture for each inversion model, and the absolute errors between the predicted values and the true values.

Figure 22 shows a statistical histogram of the frequency of absolute errors in soil moisture accounted for by the three model inversions at station GP02. As shown in the figure, the absolute error distribution of the linear model is between 0.5 and 1.5, the BP neural network model has an absolute error distribution of −0.5 to 0.5, the GABP neural network model has an absolute error distribution of −0.25 to 0.25, and overall, the three models conform to a normal distribution.

**Table 10.** Predicted soil moisture values for each model.

| Monitoring Station | Actual Soil Moisture Value/% | | Model Predicted Soil Moisture Values/% | | | | Absolute Error | | | |
|---|---|---|---|---|---|---|---|---|---|---|
| | | | Linear | | BP | GABP | Linear | | BP | GABP |
| | | | amp | Phase | | | amp | Phase | | |
| | max | 35.0 | 31.0 | 33.0 | 32.5 | 32.6 | −4.0 | −2.0 | −2.5 | −2.4 |
| GP01 | median | 28.5 | 28.3 | 27.8 | 28.5 | 28.3 | −0.2 | −0.7 | 0.0 | −0.2 |
| | min | 24.8 | 25.6 | 25.1 | 25.8 | 25.6 | 0.8 | 0.3 | 1.0 | 0.6 |
| | max | 28.5 | 27.7 | 28.3 | 27.7 | 28.5 | −0.8 | −0.2 | −0.8 | 0.0 |
| GP02 | median | 26.9 | 26.7 | 27.0 | 26.7 | 27.3 | −0.2 | 0.1 | −0.2 | 0.4 |
| | min | 25.1 | 24.7 | 25.4 | 25.3 | 25.1 | −0.4 | 0.3 | 0.2 | 0.0 |
| | max | 29.0 | 28.5 | 28.8 | 28.0 | 28.6 | −0.5 | −0.2 | −1.0 | −0.4 |
| GP03 | median | 28.0 | 27.7 | 27.6 | 27.5 | 28.0 | −0.3 | −0.4 | −0.5 | 0.0 |
| | min | 27.5 | 28.0 | 28.1 | 27.0 | 27.5 | 0.5 | 0.6 | −0.5 | 0.0 |

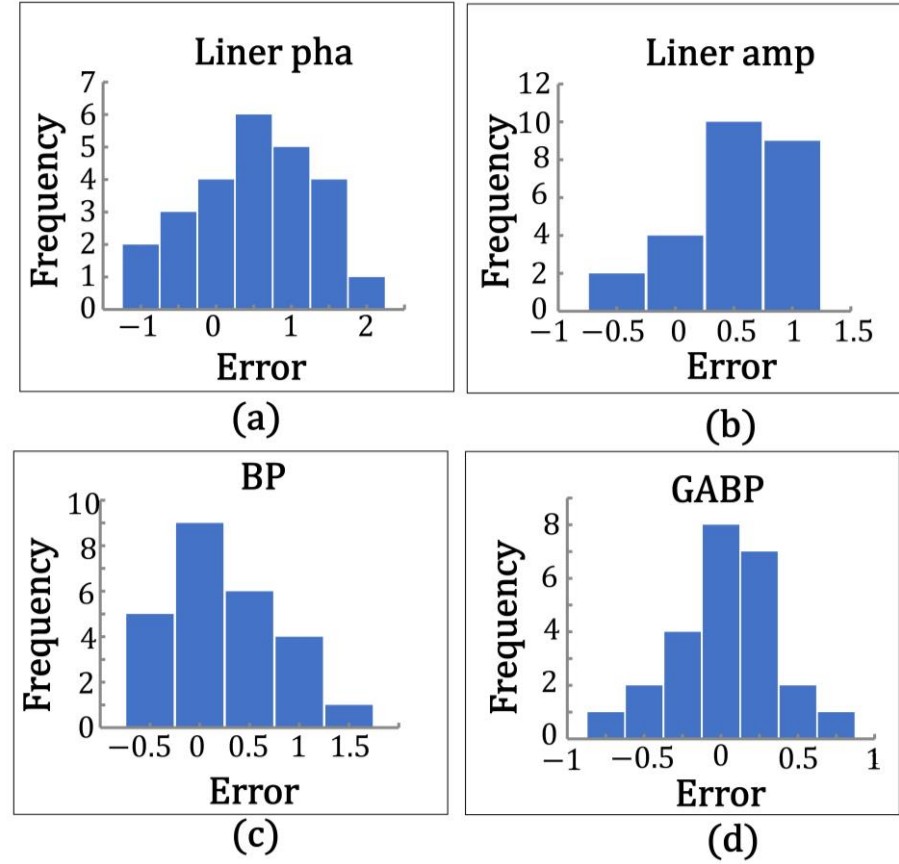

**Figure 22.** Absolute error distribution map of various models at GP02 Station. (**a**) Distribution of absolute errors in the linear model of amplitude. (**b**) Distribution of absolute errors in the linear model of phase. (**c**) Absolute error distribution of BP neural network models. (**d**) Absolute error distribution of GABP neural network models.

From the above experimental results, it can be seen that the accuracy of the linear model, BP neural network model, and GA-BP neural network model built by fusing multi-satellite multi-band data by using the technique proposed in this study was higher than that of the model built from single-satellite data, which fully proved the feasibility and effectiveness of the method proposed in the study. The values predicted with the GA-BP neural network model were closer to the true values received by the sensors. Figure 23 shows the plot of soil moisture values and error in the true values for the inversion of

the three models, where yellow is the predicted value and error in the true values for the GA-BP neural network model, green is the predicted value and error in the true values for the BP neural network model, red is the predicted value and error in the true values for the linear model built with the amplitude of the characteristic elements, and blue is the predicted value and error in the true values for the linear model built with the phase of the characteristic elements.

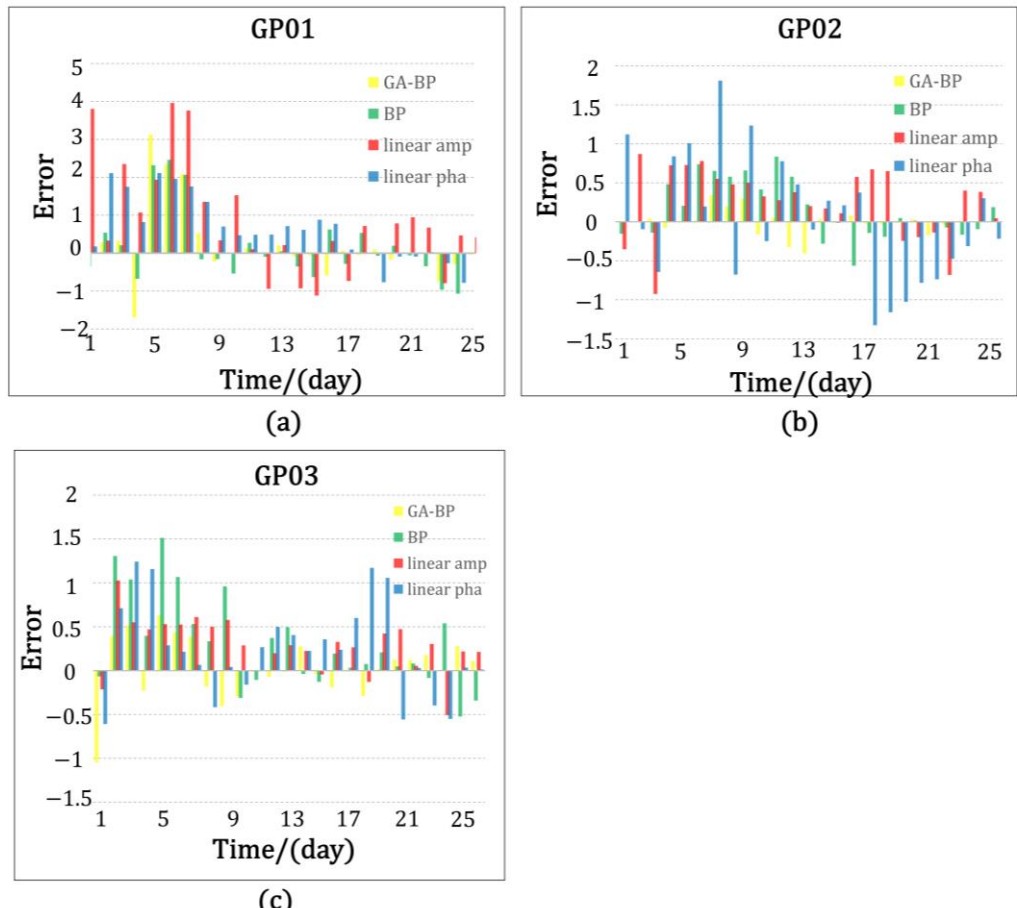

**Figure 23.** Chart of the inverted soil moisture values and error in the true values. (**a**) The error between the predicted and true values of soil moisture for each model at GP01 station. (**b**) The error between the predicted and true values of soil moisture for each model at GP02 station. (**c**) The error between the predicted and true values of soil moisture for each model at GP03 station.

## 6. Conclusions

In this study, based on GNSS-R technology and deep learning method, we carried out a study on soil moisture inversion for the channel slope of the deep excavated expansive soil canal section of the South-to-North Water Diversion Middle Line Project in China, and provided a systematic solution for soil moisture inversion in the study area, which provided scientific data support for analyzing the deformation mechanism of the channel slope in the study area, and the main conclusions of the study are as follows:

(1) In order to improve the accuracy of soil moisture inversion by GNSS-R technology, a multi-satellite and multi-band data fusion technique is proposed based on the least squares adaptive fusion algorithm and entropy value method, which provides a solution to the problems of limited observation information and low inversion accuracy of soil moisture in single-satellite inversion. Combining the fused data with linear models, BP neural network models, and GA-BP neural network models for soil moisture inversion experiments, it can be seen that compared with single satellite retrieval, the root mean square deviation of the three models decreased by 0.893,

1.207, and 1.207, respectively, which indicates that it is feasible and reliable to use the multi-satellite multi-band data fusion technology proposed in this study to retrieve soil moisture in the study area.

(2) The inversion analysis of soil moisture near the three GNSS stations was carried out using linear model, BP neural network model, and GA-BP neural network model, respectively, and the results showed that the results of inversion of soil moisture using GA-BP neural network were better than the other two models, and the correlation degree of the three sites is as low as 84.8% and as high as 95.4%, which indicates that the comprehensive use of multi-satellite multi-band data fusion technology and GA-BP neural network model inversion of soil moisture can achieve good results. It provides a new technical path for the soil moisture inversion of deep excavated expansive soil channel slopes in the South-to-North Water Diversion Project.

(3) The GNSS-R soil moisture inversion process is affected by terrain conditions and soil roughness. The application scenario of this paper is the slope of the channel of the South-to-North Water Diversion Middle Line Project, and the study area has a low vegetation coverage, so the influence of vegetation on soil moisture is not considered. In the future, we will further optimize the soil moisture inversion model based on GNSS-R and deep learning, focusing on the influence of vegetation on the inversion results, to achieve a more realistic soil moisture inversion and to expand the application scenarios of the research results.

**Author Contributions:** Conceptualization, Q.H., Y.L., W.L. (Wenkai Liu) and X.L.; methodology, Q.H., P.H. and X.L.; software, Y.L., P.W., W.L. (Weiqiang Lu) and Y.K.; validation, Q.H., K.M. and D.Z.; formal analysis, Q.H.; investigation, Q.H.; resources, Q.H., K.M. and D.Z.; data curation, H.H. and Y.L.; writing—original draft preparation, Q.H., D.Z., P.W. and Y.L.; writing—review and editing, K.M. and P.H.; supervision, Q.H. and W.L. (Wenkai Liu); project administration, Q.H., X.L. and W.L. (Wenkai Liu); funding acquisition, Q.H., X.L. and W.L. (Wenkai Liu). All authors have read and agreed to the published version of the manuscript.

**Funding:** This research was funded by the National Natural Science Foundation of China (Nos. 42277478, 52274169, and 41301598) and the Joint Funds of the National Natural Science Foundation of China (No. U21A20109). The authors would like to thank the editor and reviewers for their contributions to the paper.

**Data Availability Statement:** Not applicable.

**Conflicts of Interest:** The authors declare no conflict of interest.

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
