# Peer review of "Research on Soil Moisture Inversion Method for Canal Slope of the Middle Route Project of the South to North Water Transfer Based on GNSS-R and Deep Learning"

_remotesensing, doi:10.3390/rs15174340_

Round 1
Reviewer 1 Report
The soil moisture from the South-to-North Water Diversion Middle Route Project is assessed in the study. Complex and variable geological conditions make difficult the prediction of soil moisture in the study area. For this aim, GNSS monitoring stations are used. Some suggestions and comments to the authors are presented below:
1. The flowchart of the suggested methodology should be given by more branches and in detail in Figures 4 & 5. Thus, the readers can easily follow the application procedures.
2. Literature part is looking weak. Give new and last updated examples from literature about “GA-BP neural network model” and “BP neural network model” as
(2020). Comparison of GA-BP and PSO-BP neural network models with initial BP model for rainfall-induced landslides risk assessment in regional scale: a case study in Sichuan, China. Natural Hazards, 100, 173-204.
(2022). Comparison of different ANN (FFBP, GRNN, RBF) algorithms and Multiple Linear Regression for daily streamflow prediction in Kocasu River, Turkey. Fresenius Environmental Bulletin, 31(5), 4699–4708.
3. The performance metrics part is weak in the paper. More metrics can be calculated to evaluate the application results additionally MAE, RMSE, correlation and R-squared as MSE, NSE, RSR (Ratio of RMSE to the standard deviation of the observations) etc. …
4. Some statistical properties as coefficient of variation, confidence intervals, distribution characteristics, min and median, etc. of used data (e.g. soil moisture) should be given in a table.
5. Conclusions part can be improved in the paper. Here is presented in a general concept.
6. Is the used methodology in the paper valid for all areas or is there any limitation or classification for the application?
7. The resolution of the figures are very low.
Check the tenses of the sentences. There are present and past tenses in a paragraph. See the paragraph in the Abstract.
There are some crucial errors.
Keywords should be ordered A to Z.
Use passive sentences. Check the sentences started by “we”. See the lines 243, 248 …
One sentence can’t be a paragraph. See the lines 279-280 …
Reviewer 2 Report
1.The topic selection is more interesting, but the research method is not novel enough.
2.The background of the study was inadequate.
Reviewer 3 Report
This study investigates an inversion method for soil moisture in the slopes of the excavated wide soil channel of the South-North water diversion middle path project based on GNSS-R and deep learning.
The article has an interesting topic and is in the field of Remote Sens. journal.
- The first sentence of the abstract section should be transferred to materials and methods.
- Please provide contributions to the article at the end of the introduction section.
- Figures need improvement. Please revise.
- Remove the highlight of the tables.
- The results and discussion are well organized with recent studies.
- What is the limitation of this study?
Minor editing of English language required.
Round 2
Reviewer 1 Report
Thank the authors for carefully revision of the paper by answering each comment from the first round. For the final version of the paper, references and citations should be checked. For example, reference 30 must be corrected as “Burgan, H.I. (2022). Comparison of different ANN (FFBP, GRNN, RBF) algorithms and Multiple Linear Regression for daily streamflow prediction in Kocasu River, Turkey. Fresenius Environmental Bulletin, 31(5), 4699–4708.”
Then, Surnames should be small letters after the first letter in the references 12, 13, 33, 36…
Doi numbers are added for some references, but some of them are missing …
Reviewer 2 Report
Are all GNSS receiving devices sufficient for this study? Please add a description of the data acquisition device.
Reviewer 3 Report
Thank you for applying comments.
